# Network integration of multi-tumour omics data suggests novel targeting strategies

Ítalo Faria do Valle[1,2], Giulia Menichetti[3], Giorgia Simonetti [4], Samantha Bruno[4], Isabella Zironi [1], Danielle Fernandes Durso[4,5], José C.M. Mombach [6], Giovanni Martinelli[4,7], Gastone Castellani[1] & Daniel Remondini [1]

We characterize different tumour types in search for multi-tumour drug targets, in particular aiming for drug repurposing and novel drug combinations. Starting from 11 tumour types from The Cancer Genome Atlas, we obtain three clusters based on transcriptomic correlation profiles. A network-based analysis, integrating gene expression profiles and protein interactions of cancer-related genes, allows us to define three cluster-specific signatures, with genes belonging to NF-κB signaling, chromosomal instability, ubiquitin-proteasome system, DNA metabolism, and apoptosis biological processes. These signatures have been characterized by different approaches based on mutational, pharmacological and clinical evidences, demonstrating the validity of our selection. Moreover, we define new pharmacological strategies validated by in vitro experiments that show inhibition of cell growth in two tumour cell lines, with significant synergistic effect. Our study thus provides a list of genes and pathways that could possibly be used, singularly or in combination, for the design of novel treatment strategies.

[1] Department of Physics and Astronomy, University of Bologna, Viale Berti Pichat 6/2, 40127 Bologna, Italy. [2] CAPES Foundation, Ministry of Education of Brazil – Setor Bancário Norte (SBN), Quadra 2, Bloco L, Lote 06, Edifício CAPES, 70.040-031 Brasília, DF, Brazil. [3] Department of Physics, Center for Complex Network Research, Northeastern University, 177 Huntington Avenue, 02115 Boston, MA, USA. [4] Department of Experimental, Diagnostic and Specialty Medicine, University of Bologna and Institute of Hematology "L. and A. Seràgnoli", Via Massarenti 9, 40138 Bologna, Italy. [5] National Counsel of Technological and Scientific Development (CNPq), Ministry of Science Technology and Innovation (MCTI), Brasilia 70000-000 DF, Brazil. [6] Department of Physics, Universidade Federal de Santa Maria, Av. Roraima 1000, 97105-900 Santa Maria, RS, Brazil. [7] Present address: Istituto Scientifico Romagnolo per lo Studio e Cura dei Tumori (IRST) IRCCS, Via Piero Maroncelli 40, 47014 Meldola, Italy. These authors contributed equally: Ítalo Faria do Valle, Giulia Menichetti, Giorgia Simonetti. Correspondence and requests for materials should be addressed to D.R. (email: daniel.remondini@unibo.it)

High-throughput molecular profiling has changed the approach to study cancer. For decades, anatomical localization and histological features have guided the identification of cancer subtypes, but the genomic profiling of tumour samples has revealed differences and similarities that go beyond the histopathological classification. The diversity in genomic alteration patterns often stratifies tumours from the same organ or tissue, while tumours in different tissues may present similar patterns[1–3]. For example, mutational profiling of regulatory proteins shows tissue specificity, while histone modifiers can be mutated similarly across several cancer types[4]. Hoadley et al.[2] suggests that lung squamous, head and neck, and a subset of bladder cancers form a unique cancer category typified by specific alterations, while copy number, protein expression, somatic mutations and activated pathways divide bladder cancer into different subtypes. The analysis of cancer transcriptomes revealed that the same tumour may originate from several cell types, and different biological processes may lead to malignant transformation[4]. Moreover, similar pathways may be activated in different cancers, like ovarian, endometrial and basal-like breast carcinomas[5,6]. Notwithstanding the enormous increase of knowledge on tumour processes, a practical application of this knowledge to new treatment strategies has not advanced with the same pace. For example, common genetic alterations can predict similar responses to pharmacological therapies across multiple cancer cell lines[7–9], thus common molecular and functional profiles could enable the repurposing of therapies from one cancer to another.

Several methods have been proposed and applied for the analysis of omics data in cancer[10]. Generally, they refer to: (a) reconstruction of regulatory networks from expression data;[11] (b) identification of network modules by clustering or network diffusion techniques (usually starting from an a priori selection of seed genes as somatic mutations and differentially expressed genes);[12–15] and (c) evaluation of cancer alterations at the pathway-level comparing many samples[16,17]. However, the search for new drug targeting and repurposing strategies requires different network approaches able to evaluate a broad list of genes and identify their individual impact in the underlying regulatory networks of several tumour types at the same time.

For this aim, we propose a study of gene networks based on expression profiling and interactome topology, in combination with cancer-specific functional annotation.

Starting from whole-genome transcriptional profiling extracted from The Cancer Genome Atlas (TCGA) data portal, we selected a curated subset of 760 cancer-related genes described both in the Ontocancro database[18], and in the BioPlex protein–protein interaction—PPI-network[19,20]. We defined three tumour clusters starting from the gene–gene correlation matrices of each tumour (see Methods). Then we performed a topological analysis of the corresponding networks based on node centrality, obtaining specific signatures for multi-tumour drug targeting and survival prognosis. The validation of our signatures through literature interrogation, clinical information and by in vitro testing, makes us confident that this study can help both clinical and research communities, providing novel targets for multi-drug approaches and for repurposing of existing drugs.

## Results

**Identification of multi-tumour gene signatures**. We analyzed transcriptomics data of 2378 samples from 11 tumour types (Supplementary Table 1) considering 760 cancer-related genes with both oncogenic and PPI annotation (Bioplex-Ontocancro network, see Methods). The tumour datasets were clustered in three groups based on their gene–gene correlation matrices (see Methods) containing,

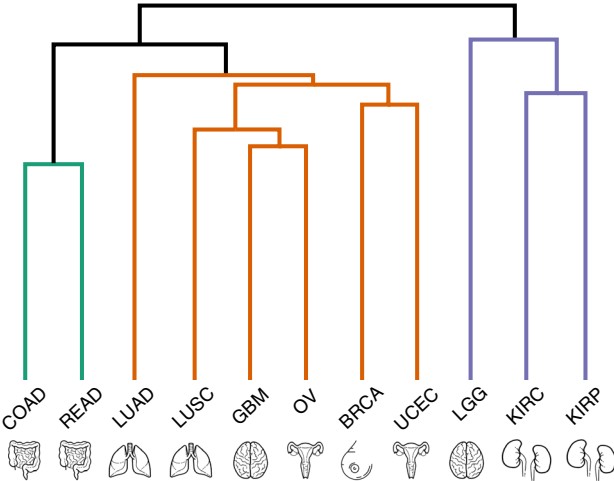

**Fig. 1** Tumour clustering. Tumour dendrogram according to the hierarchical clustering of 760-gene z-scores filtered by CLR algorithm. Brain, lung, intestine, uterus, and kidneys icons made by Kirill Kazacheck (https://www.flaticon.com/authors/kirill-kazachek) and breast icon made by Freepik (https://www.flaticon.com/authors/freepik)

respectively, 2, 6 and 3 cancer types: (1) Colon adenocarcinoma (COAD) and Rectum Adenocarcinoma (READ); (2) Lung Adenocarcinoma (LUAD), Lung Squamous Cell Carcinoma (LUSC), Glioblastoma Multiforme (GBM), Ovarian Serous Cystadenocarcinoma (OV), Breast Invasive Carcinoma (BRCA), and Uterine Corpus Endometrial Carcinoma (UCEC); and (3) Brain Lower Grade Glioma (LGG), Kidney Renal Clear Cell Carcinoma (KIRC), and Kidney Renal Papillary Cell Carcinoma (KIRP) (Fig. 1). We superimposed the gene–gene correlation matrices (calculated with the samples of all tumours inside each cluster) onto the BioPlex-Ontocancro network: a network of genes resulting from the intersection of the BioPlex network (containing physical protein–protein interactions measured via mass spectrometry) with the Ontocancro database (with annotations for cancer-related genes and pathways, see Methods). We obtained three weighted networks, each with approximately 80% nodes and 60% edges of the original BioPlex-Ontocancro network (Supplementary Note 1, Supplementary Table 2, Supplementary Figures 1–4).

We hypothesized that the most central genes in each network should play a fundamental role in the tumours represented in the cluster, in accordance to similar approaches proposed in previous works[21,22].

We used Spectral Centrality (SC)[23] as centrality measure, which is related to the changes in network global diffusivity by node perturbation through a Laplacian formalism. We applied SC in the largest connected components and considered as cluster-specific signatures the nodes with SC above the 90th percentile (25, 27 and 24 genes for clusters 1, 2, 3 respectively, Table 1 and Supplementary Data 6). As the BioPlex-Ontocancro network was differentially filtered by each cluster-specific correlation matrix, we compared the resulting signatures with the most central genes in the network prior to the filtering by any correlation matrix, and observed small overlaps between them (3/25, 13/27 and 4/24 common genes for clusters 1, 2, 3, respectively) showing how the added information of gene expression profiling correlation contributes in the creation of networks that are highly specific for the tumours in each cluster (Supplementary Table 12). The top-ranking nodes also differ significantly from those obtained from other centrality measures such as degree and betweenness centrality (see Supplementary Table 3) and even if some signature genes overlap between clusters, their links are different (Fig. 2,

| Table 1 Signature genes. List of signature genes for the three tumour clusters | | | |
| --- | --- | --- | --- |
| | **Cluster 1** | **Cluster 2** | **Cluster 3** |
| Spectral centrality > 90th percentile | *ALOX5* | *BTRC* | *AKT2* |
| | *BTRC* | *CENPC1* | *ALOX5* |
| | *BUB1* | *CETN2* | *BAG4* |
| | *CDC20* | *DSN1* | *CAPN1* |
| | *CENPC1* | *ERCC1* | *CAPN2* |
| | *CHUK* | *ERCC4* | *CDC16* |
| | *CUL1* | *FANCB* | *CDC27* |
| | *MIS12* | *FYN* | *CDT1* |
| | *MLF1IP* | *H2AFX* | *ENDOG* |
| | *NDC80* | *IL6R* | *FBXW11* |
| | *NFKB1* | *MCM10* | *FNTA* |
| | *NFKB2* | *MIS12* | *GMNN* |
| | *NFKBIA* | *MLF1IP* | *KIF2B* |
| | *PMF1* | *NEDD1* | *KIF2C* |
| | *PPP2CB* | *NFKB1* | *LSP1* |
| | *PPP2R5D* | *NFKBIA* | *NEDD1* |
| | *PSMB9* | *NUP43* | *PRKACG* |
| | *PSMC2* | *PARP1* | *PSMC3* |
| | *PSMF1* | *PLK1* | *PSMD9* |
| | *RAD21* | *PSMB3* | *SKP2* |
| | *REL* | *PSMC3* | *TNFRSF1A* |
| | *RELB* | *RPA2* | *TUBGCP5* |
| | *RPS27* | *SRC* | *UBB* |
| | *SRC* | *TNFRSF10B* | *VIM* |
| | *STAG1* | *TUBGCP5* | |
| | | *TUBGCP6* | |
| | | *XPA* | |

Supplementary Figures 5–6, Supplementary Note 3, Supplementary Figure 9), evidencing specific interaction patterns. We observed an overlap (>50%) of our signatures with signatures obtained with the same procedure when protein–protein interactions from different experiments[24,25] (i.e., yeast-two hybrid) were added to the BioPlex-Ontocancro network, supporting the robustness and biological relevance of the observed signatures (Supplementary Note 4, Supplementary Tables 4–5).

The signatures contain genes related to three biological categories: NF-κB signaling, ubiquitin-proteasome system and chromosomal instability, the last category referring to genes involved in kinetochore formation, microtubule dynamics and chromosome segregation functions (Table 2 and Supplementary Data 2 for the list of Gene Ontology enriched terms). The signatures have at least one substrate recognition component of E3 ubiquitin ligase complexes: *BTRC* in clusters 1 and 2; and *FBXW11* in cluster 3. Cluster 1 has genes involved in spindle checkpoint (*BUB1*, *CDC20*). The cluster 2 signature has many genes related to DNA repair (*CETN2*, *FANCB*, *H2AFX*, *ERCC1*, *ERCC4*, *PARP1*, *XPA*) and DNA replication (*RPA2*, *MCM10*). Moreover, it has three important genes in the signaling path that activates the *STAT3* transcription factor: *SRC*, *NFKB1* and *IL6R*. Indeed, the *STAT3* gene expression levels are significantly higher in cluster 2 (one-way ANOVA $p$-value: $5.58 \times 10^{-15}$) both in comparison with cluster 1 and cluster 3 (one-way ANOVA Student's $t$ test $p$-values $1.08 \times 10^{-9}$ and $1.14 \times 10^{-8}$, Supplementary Figure 10). The cluster 3 signature contains genes involved in three different apoptotic mechanisms: induced by TNF-α (*TNFRSF1A* and *BAG4*), induced by endoplasmatic reticulum stress (*CAPN1* and *CAPN2*) and caspase-independent apoptosis (*ENDOG*).

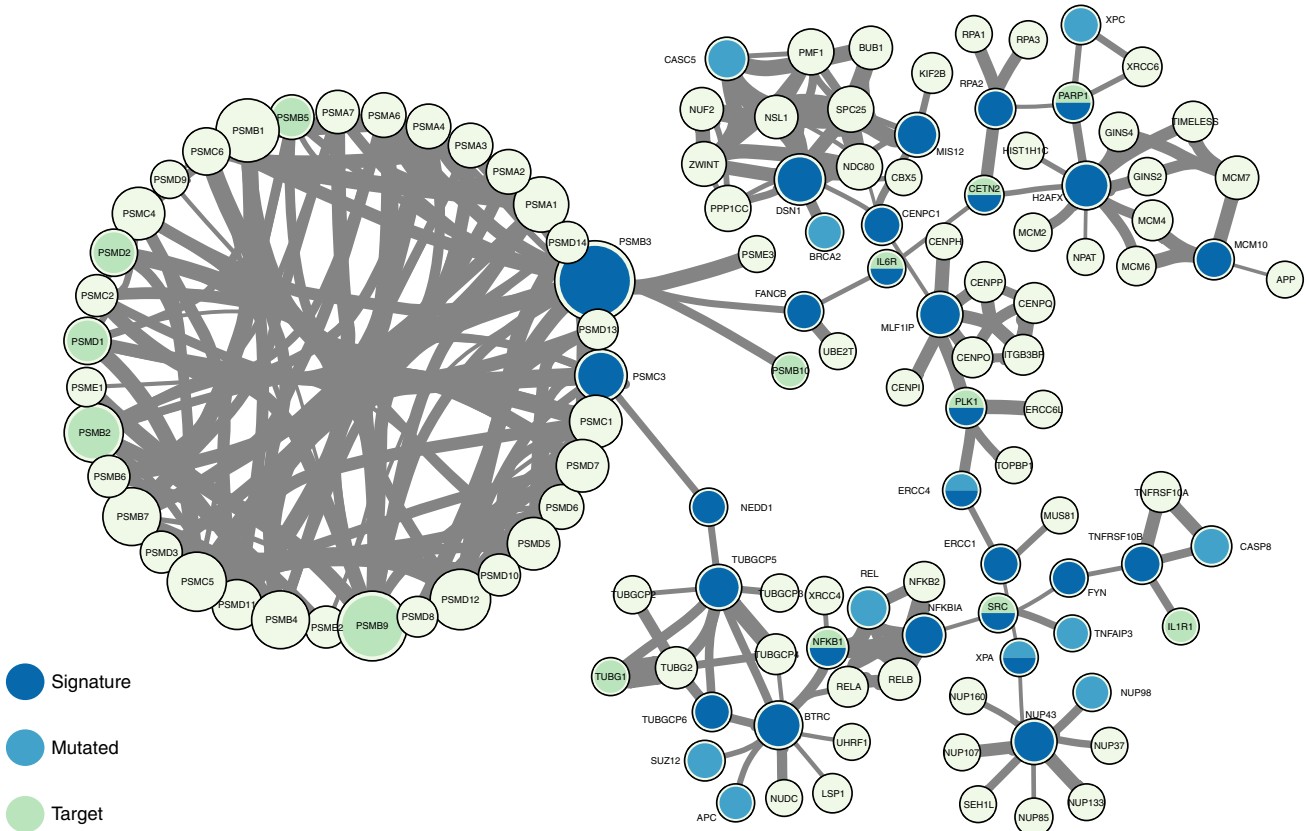

**Fig. 2** Network context of cluster 2 signature genes. Network composed by the first neighbours of cluster 2 signature genes in the BioPlex-Ontocanco network. Node sizes are proportional to their degree in the network and edge thickness is proportional to the normalized (CLR) co-expression between genes

**Table 2 Common biological categories in gene signatures**

|  | NF-κB signaling | Chromosomal instability | Ubiquitin-proteasome system |
|---|---|---|---|
| Cluster 1 | BTRC, CUL1, SRC, NFKBIA, NFKB1, NFKB2, REL, RELB, CHUK | CDC20, BUB1, MLFPIP, CENPC1, MIS12, PMF1, NDC80, RAD21, STAG1 | BTRC, CUL1, PSMB9, PSMC2, PSMF1 |
| Cluster 2 | BTRC, SRC, NFKBIA, TNFRS10B, IL6R | MIS12, DSN1, MLFPIP, CENPC1, PLK1, NEDD1, TUBGCP5, TUBGCP6 | BTRC, PSMB3, PSMC3 |
| Cluster 3 | FBXW11, AKT2, TNFR1A | CDC16, CDC27, NEDD1, TUBGCP5, KIF2B, KIF2C | FBXW11, PSMC3, PSMD9 |

All signatures have genes that can be grouped into the following categories: NF-κB signaling, chromosomal instability and ubiquitin-proteasome system

Then, we searched for possible relationships between the signature genes and genes commonly mutated across patients (Supplementary Note 4, Supplementary Tables 4–5). We observed that some signature genes also present somatic mutations (REL and RAD21 in cluster 1, ERCC4 and XPA in cluster 2, and AKT2 in cluster 3) or that mutated genes are direct neighbours of the signature genes in the network (Fig. 2, Supplementary Figures 2–6). A permutation test over the signature labels reveals a significant proximity of signature genes to mutated genes for cluster 1 and cluster 2 (random permutation test $p$-value = $8.76 \times 10^{-4}$ and $p$-value = $6.9 \times 10^{-3}$ respectively, Supplementary Figure 11). For the particular case of cluster 3, only one mutated gene is present in the signature neighbourhood and it is successfully selected as a signature gene (Fig. 2). These outcomes highlight the strict relationship between signature genes and key processes in tumour development (in analogy with the network-based approach of Novarino et al.[26]).

Since the signature genes are the most central nodes in the three tumour cluster networks, we hypothesized that they might be suitable drug targets. For this purpose, we collected the drugs that target genes in the signatures (Supplementary Data 4) and we evaluated in the ClinicalTrials repository if these drugs were under ongoing clinical trials for cancer treatment. We observed that 11 genes from the cluster signatures are being tested: four and three genes, from cluster 1 and 2, respectively; three genes from both cluster 1 and 2; and one gene from both cluster 1 and 3 (Table 3, Supplementary Data 5). Interestingly, by considering all drugs that target genes in the BioPlex-Ontocancro network, we observed a significant overrepresentation of drugs that target genes in cluster 2 gene signature (Supplementary Note 8, Supplementary Table 9).

**Gene signatures differentiate survival outcome**. We then asked whether the expression levels of the signature genes could predict patient survival in each cluster, independently of the tumour type. For cluster 1 and 3, survival information was available only for 17 and 32 patients, respectively, which resulted in non-significantly different survival curves, possibly due to the low power of the test (Supplementary Note 9, Supplementary Figures 14–15). For cluster 2, we retrieved the clinical information for 1884 patients: the survival curves showed that the 27-gene signature significantly separated the patients in two groups according to good or bad survival outcome (Log-rank test $p$-value = $7.26 \times 10^{-18}$, Fig. 3a). We tested the significance of this separation against randomly generated signatures of the same size. Since one of the main factors affecting signature performance was the imbalance of samples from the same tumour in the two groups, we introduced a measure of sample imbalance to be considered together with the $p$-value (see Supplementary Figure 16). Our results show that our gene signature is quite balanced, and that it outperforms random signatures with similar extent of sample imbalance (Supplementary Figure 16). Although confounding factors may affect the performance, a multivariate analysis that considered patient age and tumour type as covariates, confirmed that the

**Table 3 Clinical trials**

| Inhibition target | Number of clinical trials | Cluster signature |
|---|---|---|
| ALOX5 | 18 | 1, 3 |
| CHUK | 9 | 1 |
| FYN | 97 | 2 |
| IL6R | 2 | 2 |
| NFKB1 | 40 | 1, 2 |
| NKFB2 | 8 | 1 |
| NKFBIA | 8 | 1, 2 |
| PARP1 | 106 | 2 |
| PPP2CB | 5 | 1 |
| PSMB9 | 25 | 1 |
| SRC | 135 | 1, 2 |

List of signature genes that are also being tested in ongoing clinical trials (according to AACT database)

separation of samples due to the gene signature played the major role in the observed difference of survival outcomes (Supplementary Data 1). When we further stratified the analysis at a single tumour level (considering its stage or grade when available) the separation resulted significant only for a set of tumours, with differences possibly due also to the different number of samples in each stage or grade considered (see Supplementary Data 1).

**Experimental validation of cluster 2 signature**. We tried to translate our results into novel therapeutic strategies by applying, for a subset of tumours in cluster 2 (which contained the largest and most heterogeneous set of tumours), a set of drugs on targets taken from the signature or from related biological categories. We selected three drugs: BI6727, for targeting the cluster 2 signature gene PLK1; Bortezomib, for targeting proteasome and NF-κB pathway; and the PF-00477736 drug, to target the CHK1/2 genes, which play a role in DNA damage response but are not in the signature. The choice was based on signature gene list (considered as equally relevant, since many factors could affect exact SC node ranking), drug availability and required experimental settings, in order to avoid potential culture biases (e.g. the need of adding specific compounds to the medium or co-culturing with other cells to test some drugs). We tested these drugs, alone or in combination, on T98G glioblastoma cell line (modelling a clinically very aggressive tumour) and MCF-7 breast adeno-carcinoma cell line (modelling a very common tumour type, see Methods for details on the experiments). Both cell lines resulted highly sensitive to Bortezomib, with an IC50 of 200 nM for MCF-7 and 0.6 nM for T98G (Fig. 4). BI6727 treatment reduced viability in a concentration-dependent manner in both models, with the glioblastoma model showing increased responsiveness (IC50 of 69.2 nM versus 1.8 μM for MCF-7). Moreover, both cell lines showed low response to PF-00477736, with IC50 of 26.9 μM for MCF-7 and 15.1 μM for T98G. We then asked whether these

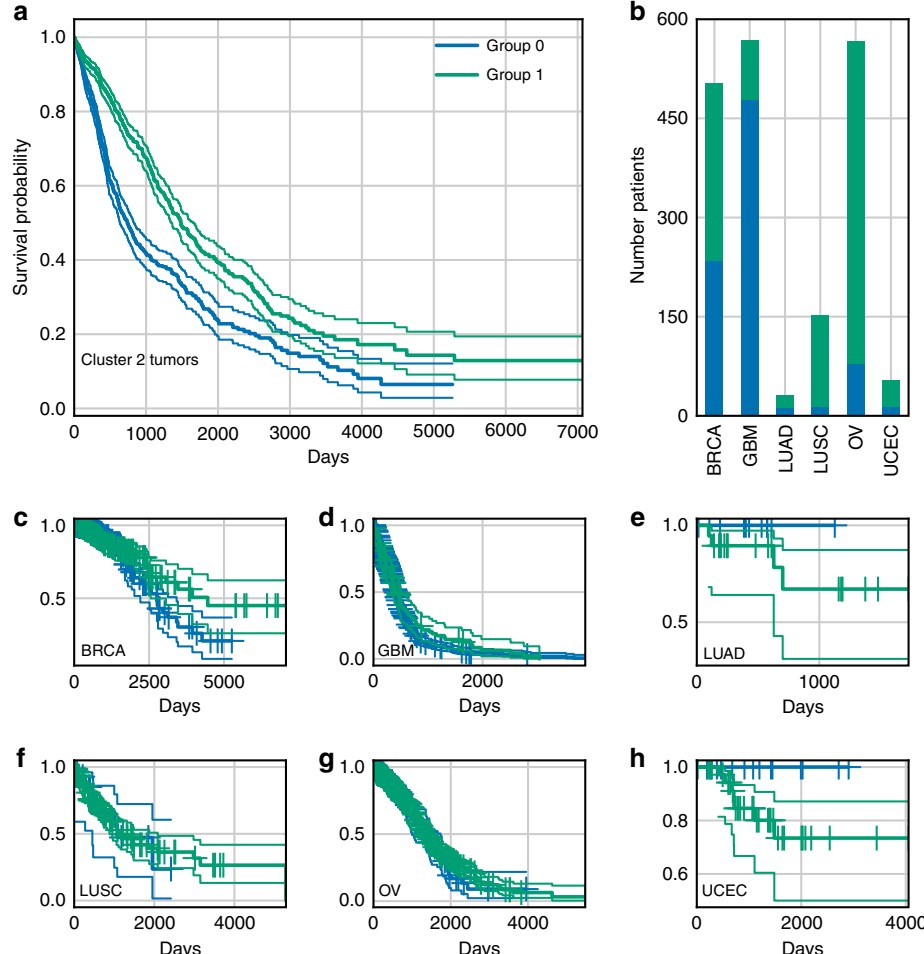

**Fig. 3** Gene signature and survival outcome. Kaplan–Meier survival curves for two groups of patients obtained via k-mean clustering of expression levels of cluster 2 signature genes. **a** Plot of the survival curves for worse (group 0) and better (group 1) outcome patient groups of all cluster 2 tumours, with a 95% confidence interval (group 0: 790 observations, 362 censored; group 1: 1094 observations, 648 censored; log-rank test $p$-value: $7.26 \times 10^{-18}$). **b** Distribution of group 0 and group 1 patients inside cluster 2 tumours, showing a fairly homogeneous distribution of samples (see Supplementary Table 10 for further discussion). **c–h** Kaplan–Meier survival curves for groups 0 and 1 considering each tumour separately

drugs might synergize in the selected models. Although the combinations of PF-00477736 with either BI6727 or Bortezomib did not show any additive or synergistic effect in both cell lines, we observed a cooperation effect between BI6727 and Bortezomib (Fig. 5a, b). Indeed, we observed that cell viability was significantly lower compared with single agent treatments in MCF-7 cells (Fig. 5a, two-sided Student's $t$ test $p$-value < 0.05), showing a general additive effect (Supplementary Data 7). We observed low Combination Index values (<1) for both cell lines, indicating synergistic effect for all concentrations tested in MCF-7, and for selected concentrations in TG98 (Fig. 5, Supplementary Data 7).

Moreover, we integrated our experimental validation results with a wider set of experimental outcomes available from the Genomics of Drug Sensitivity in Cancer (GDSC) project[27], which reports the drug screening experiments for 224,510 drug-cell line pairs (265 drugs, 1074 cell lines) (Supplementary Note 10). We observed that the IC50 values obtained in our experimental validation are lower that the majority of those reported in the GDSC project, being in the 1st, 6th, 10th, and 24th percentiles of the IC50 distribution for all drug-cell lines pairs (Supplementary Figure 17) and in the 1st, 9th, 7th, and 22nd percentiles of the IC50 distribution of all drugs specifically tested on MCF-7 and T98G cell lines (Supplementary Figure 18).

**Cell line sensitivity to drugs targeting signature genes**. We also asked whether signature genes in each cluster could predict a better response of cell lines to the related drugs when compared with all drugs tested in GDSC. We identified 56, 218, and 49 cell lines associated to the tumour types grouped in cluster 1, 2, and 3, respectively (according to TCGA classification, Supplementary Data 8). We identified 7, 15, and 7 drugs targeting genes belonging to signature 1, 2, and 3, respectively. We observed that most of the cell lines in each cluster (51/53, 103/218, and 47/49, respectively) were more sensitive to drugs targeting signature genes as compared with all other drugs (according to IC50 values, Supplementary Figures 19–21). Specifically, we found that 4, 18, and 10 cell lines in clusters 1, 2, and 3, respectively, showed significant differences (two-sided Student's $t$ test $p$-value < 0.05, Fig. 6), all of them displaying lower IC50 value when treated with drugs targeting signature genes. Remarkably, no significant differences with the opposite trend (higher IC50 values) were found. The trend remained unaltered when the IC50 values were compared between drugs targeting signature genes versus drugs targeting non-signature genes that belonged to the BioPlex-Ontocancro network: 50/53, 63/218, and 40/49 cell lines presented lower IC50 values and for 2, 6, and 4 cell lines in clusters 1, 2, and 3, respectively, the difference

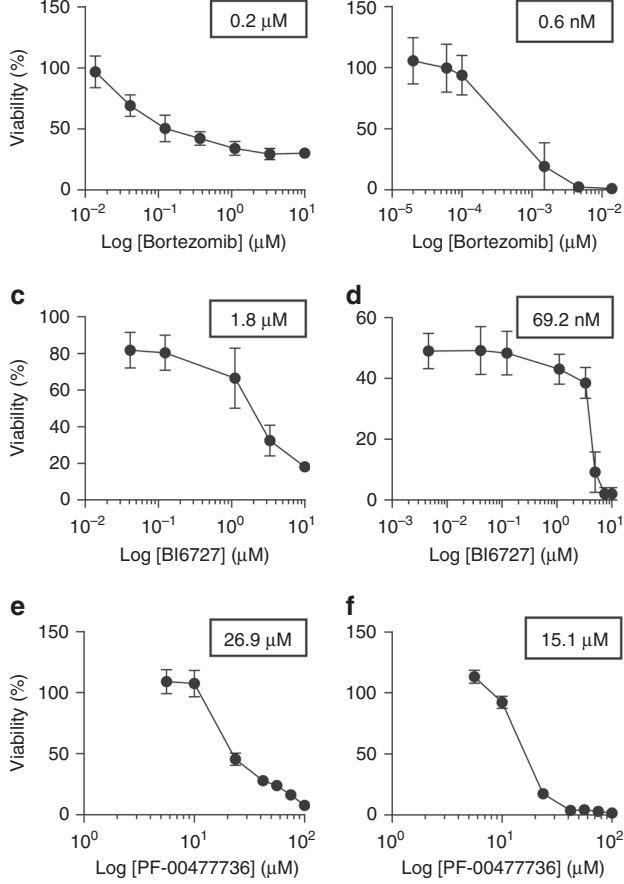

**Fig. 4** Cluster 2 cell lines response to Bortezomib, BI6727 and PF-00477736. MCF-7 and T98G cells were treated with increasing doses of Bortezomib (0.01–10 μM for MCF-7 (**a**), 0.02 –10 nM for T98G (**b**)), BI6727 (0.04–10 μM for MCF-7 (**c**), 0.004–10 μM for T98G (**d**), PF-00477736 (5.6–100 μM (**e**, **f**)) and cell viability was measured 72 h after drug administration by WST-1 assay (three independent experiments). Cell viability is represented as mean (error bars: s.e.m.). IC50 values are shown in the boxes

was statistically significant (two-sided Student's *t* test *p*-value < 0.05, Supplementary Figure 22).

## Discussion

We studied the expression profiles of 11 tumours by considering a selected set of genes annotated in the Ontocancro database and BioPlex PPI network. This knowledge-based selection reduced the dimensionality of the data to a highly curated list of cancer-related genes, involved in pathways that are hallmarks of cancer as cell cycle, inflammation, and apoptosis[28]. This approach also ensured that all studied genes had protein–protein interaction annotations, which are crucial to the understanding of how the signaling transduction propagates in the cell[29]. We clustered tumours by their gene–gene correlation matrices, to evaluate the functional relationships between genes and their impact on transcriptome organization. Many tumours from the same organ grouped together, in agreement with previous studies showing that tissue-of-origin features provide the dominant signals in the identification of cancer subtypes[2,30]. However, the clustering also grouped tumours originating from different tissues, according to similarities in genomic alterations, as in the case of BRCA, OV, LUSC, and UCEC, which share common characteristics as presence of *TP53* mutations and multiple recurrent chromosomal gains and

losses[3]. In particular, BRCA and UCEC grouped into a well-defined subcluster, which may reflect their better prognosis when compared to other ten tumour types[2]. Interestingly, the tumors LGG, KIRC and KIRP clustered in the same group, suggesting that these tumours might activate the same pathways and oncogenic processes. Although Hoadley et al.[2] did not evaluate the tumours LGG and KIRP, the authors observed that glioblastoma GBM and kidney cancer KIRC clustered in the same group when using gene expression and copy number alteration profiles.

We integrated different types of biological information by a network approach, that allowed us to identify functional modules and to rank genes as network elements[21,22,31,32]. We created a network for each cluster (starting from a common template of protein interactions and superimposing cluster-specific expression correlation profiles) and obtained specific gene signatures based on node ranking by centrality measures. Gene Ontology enrichment analysis resulted in pathways mainly associated to NF-κB signaling, chromosomal instability and ubiquitin-proteasome system. NF-κB signaling regulates genes that participate in cell proliferation, innate and adaptive immune responses, inflammation, cell migration, and apoptosis regulation processes. The aberrant activity of NF-κB may act as survival factor for transformed cells which would otherwise become senescent or apoptotic[33]. Chromosomal instability category involves kinetochore formation, microtubule dynamics and chromosome segregation functions. The dysfunction in these genes may cause cell inability to faithfully segregate chromosomes, generating genomic alterations as DNA mutation, chromosomal translocation, and gene amplification. The mutant genotypes may confer beneficial phenotypic traits to cancer cells, such as sustained proliferative signaling and resistance to cell death[28]. Two genes classified into this category have already been related to clinical practice: the prognostic marker *KIF2C*;[34,35] and the *BUB1* gene, whose expression correlates with poor clinical diagnosis[36,37]. The ubiquitin-proteasome system is the major degradation machinery that controls the abundance of critical regulatory proteins. Perturbation of the regulatory proteins turnover disturbs the intricate balance of signaling pathways and the cellular homoeostasis, contributing to the multi-step process of malignant transformation[38]. Proteasome inhibitors have become valuable tools in the treatment of certain types of cancer, mainly because malignant cells show greater sensitivity to the cytotoxic effects of proteasome inhibition than non-cancer cells[39].

In addition to common features, cluster 2 signature has several genes related to DNA repair (*CETN2*, *FANCB*, *H2AFX*, *ERCC1*, *ERCC4*, *PARP1*, *XPA*) and DNA replication (*RPA2*, *MCM10*). Interestingly, the tumours in this cluster usually present high rates (50–90%) of samples with mutated *TP53*, which is an important sensor for the cell DNA damage response[2,4,40]. The cluster 2 signature also presents the *SRC*, *NFKB*, and *IL6R* genes, which participate in the activation of *STAT3*, a transcription factor that is necessary for cell transformation[41]. We observed that *STAT3* gene expression is higher in the tumours of cluster 2 when compared with the tumours of clusters 1 and 3 (one-way ANOVA *p*-value: $5.58 \times 10^{-15}$, see Supplementary Figure 10). The cluster 3 signature has genes involved in three apoptotic mechanisms, which are induced by TNF-α (*TNFRSF1A* and *BAG4*), or endoplasmatic reticulum stress (*CAPN1* and *CAPN2*) and caspase-independent apoptosis (*ENDOG*). As the regulation of cell death serves as a natural barrier to cancer development, these processes may reflect different strategies that these tumours use in response to various cellular stresses. We remark that different experimental approaches for protein–protein interaction (e.g. yeast two-hybrid and affinity-purification followed by mass spectrometry) usually present low overlap, mainly due to assay complementarity, and differences in sensitivity and search

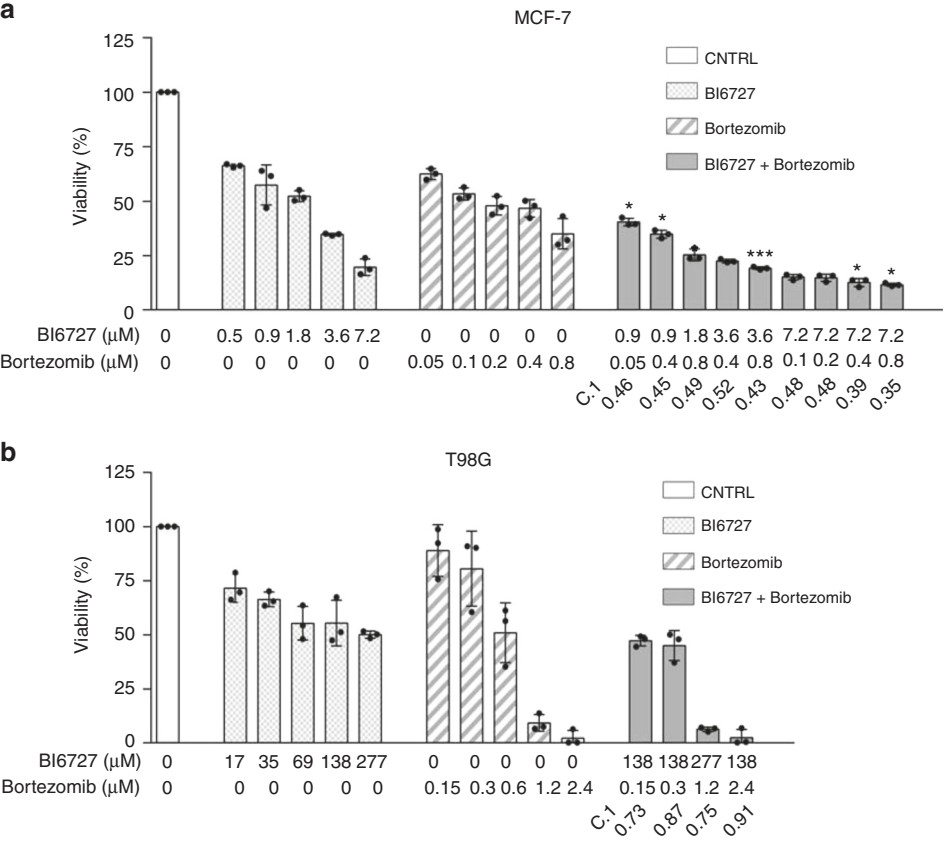

**Fig. 5** Combined inhibition of PLK1 and proteasome in MCF-7 and T98G cells. MCF-7 (**a**) and T98G (**b**) cells were treated with increasing doses of Bortezomib (0.05–0.8 μM for MCF-7, 0.15–2.4 nM for T98G) and BI6727 (0.5–7.2 μM for MCF-7, 17–277 nM for T98G), alone or in combination, and cell viability was measured 72 h after drug administration by WST-1 assay (three independent experiments). Cell viability is represented as mean (error bars: s. d.). Statistical significance was determined by two-sided Student's *t* test (*P < 0.05; ***P < 0.001). C.I. Combination Index (CompuSyn software). **a** MCF-7 cells: combinations with a C.I. lower than 0.5 are shown. **b** T98G cells: combinations showing synergistic effect are shown

space[42]. Consequently, the found gene signatures might be dependent on the protein network used in this study, despite the fact that we observed a >50% overlap in all tumour clusters when protein–protein interactions obtained from different experimental approaches were included in the BioPlex-Ontocancro network to generate different signatures (Supplementary Table 4).

Since the transcriptional disturbances observed in cancer can sometimes be explained by underlying somatic mutations[43,44] we retrieved TCGA mutational data, and focused on cancer-related mutations reported in the Catalogue of Somatic Mutations in Cancer (COSMIC) database. Many signature genes resulted also somatically mutated, or first neighbours to mutated genes (Fig. 2; Supplementary Figures 5–6, Supplementary Figures 11–13, and Supplementary Table 8), showing their strict relationship and the functional relevance of the biologically processes they are involved in.

In addition, several genes in the signatures or in their direct network neighbourhood are already under clinical investigation in a variety of tumour conditions (as annotated in Clinicaltrials. org database). For example, the *AKT* pathway has been described as a potential drug intervention in clear cell renal carcinoma:[45] *AKT2* gene belongs to the signature of cluster 3 (comprising LGG, KIRC, and KIRP), it is somatically mutated in the tumours of cluster 3 and it has been annotated as drug-target according to the DrugBank database.

We asked whether gene signatures could predict survival outcomes in each cluster, thus independently on the single tumour type. Our results show that in cluster 2 (the only one with enough available samples) the 27-gene signature defined two groups of patients with significantly different Kaplan–Meier survival curves,

also in comparison with randomly generated signatures (log-rank test *p*-value: $7.26 \times 10^{-18}$, see Supplementary Figure 16). However, we remark that the clinical meaning of this separation requires further investigation, considering the high correlation of gene expression profiles in tumours[46,47] and since differences exist at the level of single tumour due to their stage or grade (see Fig. 3, Supplementary Table 11, Supplementary Data 1). We tested three existing drugs (two targeting elements of cluster 2 signature, and one involved in a related biological process, but not directly belonging to the signature) on two tumour types of the cluster, a very aggressive one (GBM, T98G cell line) and a common one (breast cancer, MCF-7 model). PF-00477736 drug (a *CHK1/2* inhibitor, not in the signature)[48] had poor effect on both cell lines. On the contrary, they resulted highly sensitive to the combination of BI6727 (an inhibitor of the signature gene *PLK1*[49]) and Bortezomib (proteasome activity inhibitor[50]). A number of tumour models is sensitive to proteasome and *PLK1* inhibitors. Indeed, genes belonging to the ubiquitin-proteasome system can be found in all clusters and *PLK1* expression is higher in cluster 2 tumours compared to the others. However, many cell lines have been reported throughout literature to be insensitive to these drugs[51–56]. Moreover, we observed a significant synergic action of Bortezomib and *PLK1* inhibitors at several dosages on both cell lines, not found when BI6727 was considered in combination with the other two signature-related drugs.

Finally, we performed in silico experiments on a broader range of cell lines and drugs by exploiting the drug screening data of the GDSC project. We observed that our signature genes predict high sensitivity to the related targeted agents on most cell lines in

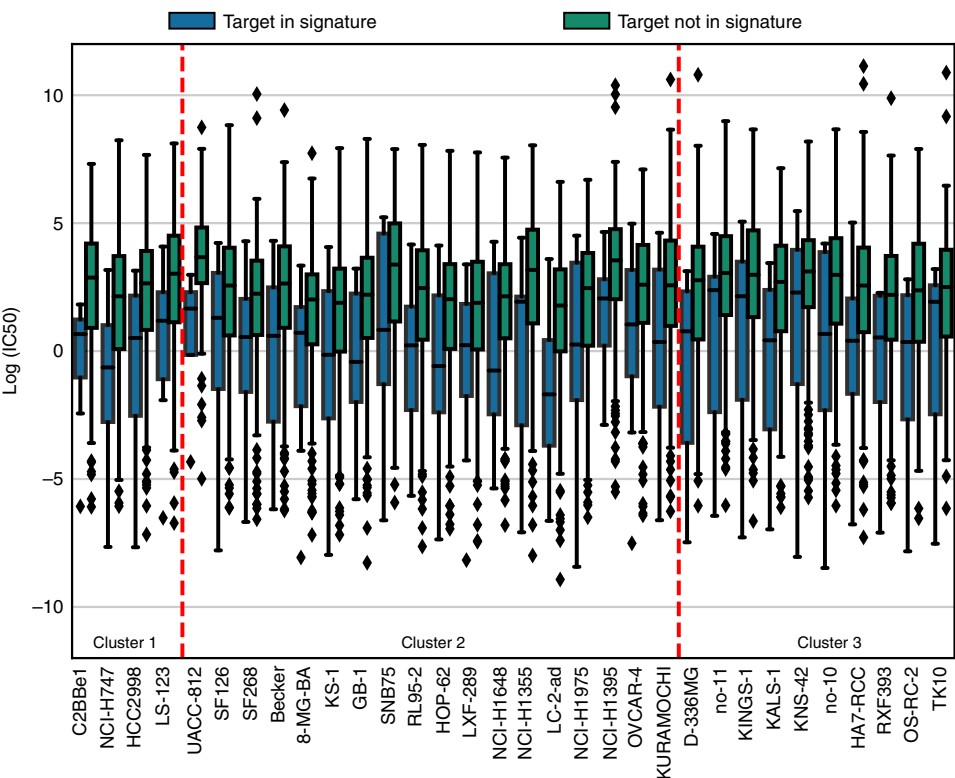

**Fig. 6** Sensitivity to drugs targeting signature genes. We retrieved drug screening data from the GDSC project and considered the cell lines associated to the tumours in our clusters. The significance was evaluated with Student's $t$ test (two-sided $p$-value < 0.05). The continuous horizontal line is the median, the lower and upper boundary represents the 25th and 75th percentiles, respectively. Whiskers extend to data points that lie within 1.5 interquartile ranges of the lower and upper quartiles; and observations that fall outside this range are displayed independently. Blue boxplots: IC50 values for signature genes; green boxplots: IC50 values for all other genes in GDSC targeting the same cell line

GDSC database that could be associated to our tumour clusters. Taken together, these findings suggest novel potential therapeutic strategies to be further explored in preclinical models.

These observations indicate that our study succeeded in: (1) clustering tumours highlighting common functional mechanisms related to their transcriptional profile, and (2) selecting genes with a relevant functional role in the studied tumours, thus amenable of drug targeting. The combination of these results may thus provide the rationale for choosing novel drug targets and drug combinations, or for repurposing existing drugs towards tumours of the same cluster. As a possible future direction, once obtained an enlarged list of novel and repurposed drugs, the specific transcriptional and mutational profile of single patients, prioritized onto our signatures, might suggest specific combinations of drugs for a more targeted and personalized therapeutic approach.

## Methods

**Gene expression datasets**. The gene expression datasets used in this study were retrieved from The Cancer Genome Atlas (TCGA, https://gdc-portal.nci.nih.gov) Data Portal, and included Agilent expression arrays of 2378 samples from 11 tumour types, with a different number of samples each (from 16 to 595, Supplementary Table 1). We selected for our analysis the genes from the BioPlex protein–protein interaction network[19,20] ($n = 10,961$) that were also present in the Ontocancro database http://ontocancro.inf.ufsm.br/, specific for tumour-related biological processes ($n = 1104$), resulting in a list of 760 cancer-associated genes related to specific biological functions (such as cell cycle control, DNA damage response, and inflammation).

**Tumour clustering**. For each tumour dataset, we calculated a correlation matrix containing pairwise Pearson $r_{ij}$ coefficients between the previously selected 760 genes across all samples available for the tumour. In order to eliminate non-significant correlations and indirect influences, the absolute correlation values ($|r_{ij}|$) were adjusted with the Context Likelihood of Relatedness (CLR) algorithm[57,58]

(implemented in the R/Bioconductor package 'minet'[59]) obtaining $z_{ij}$ positive and negative scores. We considered only positive values ($z_{ij} > 0$), but the obtained clusters resulted fairly robust when different threshold values were applied (Supplementary Note 2, Supplementary Figures 7–8). The filtered z-score matrices were clustered using hierarchical clustering analysis (with Ward linkage) based on the element-wise Euclidean distance between each pair of tumour matrices A and B, calculated as follows:

$$d(A, B) = \sqrt{\sum_{i=1}^{n} \sum_{j=1}^{n} \left( a_{ij} - b_{ij} \right)^2} \qquad (1)$$

where $a_{ij}$ is the z-score between the genes $i$ and $j$ in Tumour A and $b_{ij}$ the z-score between the same genes in Tumour B. The clusters were defined using a dynamic branch cutting algorithm implemented in the R package dynamicTreeCut[60] (parameters: minClusterSize = 2, cutHeight = 590 and method = 'Hybrid').

**Multi-tumour gene signatures**. A network approach was applied to find gene signatures that characterized the clusters of tumours. First, we created a template network (BioPlex-Ontocancro PPI) by selecting the genes present in the BioPlex protein–protein interaction network that were also annotated in the Ontocancro database. In the initial list of 760 genes used for tumour clustering, only 591 were connected to each other (169 isolated nodes) thus considered for network analysis. Then, for each cluster the gene–gene correlation coefficients $r_{ij}$ were computed, considering all the samples of all tumours in the cluster, and their absolute values $|r_{ij}|$ were adjusted with the CLR algorithm, producing the $z_{ij}$ scores[57,58]. Each score matrix was superimposed to the BioPlex-Ontocancro network, producing three weighted networks (one for each cluster) in which genes were linked only if having correlated expression profiles ($z_{ij} > 0$, specific to each cluster) and a physical interaction at protein level (given by Bioplex-Ontocancro network, common to all clusters). We remark that the three cluster-related networks can differ for their weight values or for missing links (due to negative z-scores set to zero). The networks were analyzed and visualized by Networkx Python package, Matlab and Cytoscape.

For the networks of clusters 1 and 3, we selected the giant components (245 and 244 nodes, respectively), and for cluster 2 we selected the two largest components, since they had similar size much larger than the other components (149 and 118 nodes) and since together resulted in a number of nodes comparable to the giant

component of cluster 1 and 3 (Supplementary Data 3). After this selection, we retrieved a gene signature for each cluster composed by the most central genes (network nodes), which were defined as those having the Spectral Centrality[23] topological measure (SC) above the 90th percentile (see Table 1, Supplementary Figures 2–6, Supplementary Data 6 with genes sorted by SC). SC calculates the effect of node removal on the network diffusivity based on the spectral properties of the Laplacian graph, and it has already been applied successfully to biological data such as the Immune System mediator network. Different results were obtained by considering Betweenness Centrality or weighted degree (Strength W) as centrality measures, as shown in Supplementary Table 3. The robustness of our signatures was tested by considering a random subsampling (50% of the initial number of patients in each cluster) for 100 times, and then counting the number of occurrences of original signature genes into the signatures obtained by applying the analysis pipeline onto the patient subsampling (see Supplementary Table 6).

**Validation of the multi-tumour gene signatures**. We evaluated the relevance of the genes in the signatures by several approaches.

First, we verified the proximity with the somatic mutational data extracted from the TCGA data portal for the considered tumours. To avoid cancer unrelated mutations, we considered only mutations that were reported also in the COSMIC database[61]. We checked whether the signature genes had been reported as somatically mutated or if they occurred in the neighbourhood of mutated genes in the networks. To quantify the proximity of gene signatures to mutated genes we located the nearest mutation (in terms of shortest paths on the network) for each signature gene, resulting in a collection of minimum distance values for each cluster, and we compared these distances with random gene signatures of the same size (Supplementary Figure 11–13, Supplementary Table 8).

Secondly, we retrieved from the DrugBank http://www.drugbank.ca/ and Drug Gene Interaction (DGIdb)[62] databases which genes in the signatures were also mapped as drug targets. Third, we checked in the Aggregate Analysis of Clinical Trials (AACT) database (https://www.ctti-clinicaltrials.org/aact-database) for the existence of ongoing clinical trials evaluating the inhibition of signature genes. Fourth, the prognostic potential of each gene signature was evaluated by considering the clinical data (days to death) available in the TCGA data portal. The patients having clinical information were clustered according to the expression levels of the gene signatures by a k-means algorithm (Python package 'scikit') considering two patient groups: good versus bad survival outcome. Survivals curves were calculated for both groups: we applied the Kaplan–Meier method with censored data and evaluated their significance with the log-rank test (Python package "lifelines"). We observed that the effect of the gene signature on dividing patients remained prevailing even when considering confounding factors such age and tumour type, and some differences emerged only when considering tumour stage or grade (see Supplementary Table 11, Supplementary Data 1). Moreover, we compared our signature with randomly generated signatures, by using two indices: $P$-value and $S_T$, a parameter indicating the imbalance of tumour samples inside both patient groups (details and results in Supplementary Figure 16). Fifth, we tested the effect of drugs inhibiting genes in our signatures or strictly related to them. The glioblastoma T98G and the breast adenocarcinoma MCF-7 cell lines were obtained from ATCC® ([lgcstandards-atcc.org]) and Leibnitz Institut DSMZ (dsmz.de), respectively. Cells were cultured at a density of $10^5$ cells/ml in RPMI medium plus 10% FBS (plus 5% Sodium orthovanadate for T98G) for 72 h with increasing concentrations of the following drugs: Bortezomib, BI6727, PF-00477736 (Selleckchem), alone or in combination. The dose range was chosen based on insights from the literature on the same cell lines or related cellular models[49,50,63–70]. One hour and thirty minutes before the end of treatment, WST-1 reagent was added to the cell medium and cell viability was measured according to manufacturer's instruction (Roche). The cell response to different drug dosages was tested in three independent experiments by duplicates of each experimental condition. The dose-effect response and the IC50 of each drug were calculated using GraphPad Prism 6 (GraphPad Software) with a 95% confidence interval. To determine synergy, combination indexes were obtained with the CompuSyn software (ComboSyn Inc.): combination index values <1, =1, and >1 indicate synergism, additive effect and antagonism, respectively (Supplementary Data 7).

Finally, we also validated the signature genes as potential drug targets by using drug screening data from the Genomics of Drug Sensitivity in Cancer (GDSC) project (http://www.cancerrxgene.org/). The drug–gene interactions were retrieved both from GDSC project and DGIdb, considering only drugs with known mechanistic actions and approved by the FDA. We associated cell lines in the GDSC project to our cluster according to TCGA tumour classification (Supplementary Figure 17–22, Supplementary Data 8).

**Code availability**. The code for the calculation of the Spectral Centrality (SC) measure is available in https://zenodo.org/record/1420602#.W6Bju5NKhBw [https://doi.org/10.5281/zenodo.1420602]. Codes used for the other analyses in this study are available from the corresponding author upon reasonable request.

## Data availability

The gene expression datasets used in this study were retrieved from The Cancer Genome Atlas (TCGA, https://gdc-portal.nci.nih.gov) Data Portal. Other data

supporting the findings of this study are available within the paper and its supplementary information files.

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

## Acknowledgements

This study was supported by the Interomics CNR Flagship Initiative; Mimomics EU FP7 Project no. 305280; NGS PTL EU FP7 Project no. 306242; the CNPq Project no. 402547/ 2012–8; the Associazione Italiana per la Ricerca sul Cancro AIRC (Investigator Grant to G. M., no. 19226); the Programma di ricerca Regione-Università 2010–2012 (L. Bolondi); and the Innovative Medicines Initiative (IMI) 2 project "HARMONY", no. 116026. We thank the Science Without Borders project of CAPES foundation (Ministry of Education of Brazil - Brasília -DF, Brazil) for the doctoral scholarship (grant number: 10186–13–1) for IFV.

## Author contributions

I.F.V., G. Menichetti and G.S. contributed equally to the paper. G.S., S.B., I.Z. and D.F.D. performed the experiments and interpreted the results. I.F.V., G. Menichetti and J.C.M. collected and analyzed the data. D.R. supervised data analysis. G.C., G. Martinelli and D. R. designed the research, interpreted the results, wrote the paper.

## Additional information

**Competing interests:** The authors declare no competing interests.

**Reprints and permission** information is available online at http://npg.nature.com/ reprintsandpermissions/

