## [Peer Review File · Nature Communications]

Reviewers' comments:

Reviewer #1 (Remarks to the Author):

This paper describes a method for combining transcriptomic data with a protein-protein interaction network and a database of cancer-related genes in order to identify gene clusters that are related to the biology of the tumour. The authors demonstrate that their method identifies known cancer related pathways, can be used to find a prognostic biomarker, and that several drugs targeted at members of the clusters have activity against cancer cell lines.

Although the subject of the study is one of intense interest to researchers working in the field of cancer genomics, I do not feel that the authors have made the case that their method represents an advance in the field. There are a very large number of network-based methods that have previously been applied to cancer (PMID: 22962493, PMID: 20529912, PMID: 22174262, PMID: 23228031, PMID: 28388297(review), PMID: 26125594(review), PMID: 28122019, PMID: 18990722, PMID: 21576238, PMID: 20169195). Most are able to identify enriched cancer-related pathways, select central genes, identify tumour subtypes and (often) identify prognostic biomarkers. The authors have not attempted to differentiate their algorithm from these earlier methods (and the many others; see the reviews for a list!)

Even on their own terms, the results presented in this paper are not overwhelming. A non-novel hierarchical clustering process (Figure 1), produces the three tumour type clusters that are examined in the rest of the paper. These clusters are very unbalanced. 85% of the tumours were placed in Cluster 2, 10% are in Cluster 1, and just 5% are in Cluster 3. Given that the vast majority of tumour types are in cluster 2, the method is not providing a finely detailed map of tumour type relatedness. Apparently all tumours of the same type clustered together, and the main surprise at this step of the analysis is that brain low grade glioma co-clustered with the kidney tumours rather than with glioblastoma multiforme. I would have been interested in some discussion of this finding.

When these transcription-based clusters are superimposed on a protein-protein interaction network it results in...three protein network clusters. Pathway overrepresentation analysis reveals three overrepresented well-known cancer-related biological processes (a non-novel finding), but the genes representing the processes are distributed relatively evenly among the three network clusters. When you examine the details of the protein-based network clusters, it emerges that there are many networks within each cluster: something on the order of 50 networks in cluster 2 (eyeballing Supplementary Figure 3). It is unclear to me what value this additional step of protein-based network clustering is adding to our understanding of the transcription-based tumour type clusters? At the least, I'd like to see some correlative data that shows that these networks have biological relevance; for example, when you select the protein-based network clusters based on the transcriptional profiles, are their nodes more connected than you would expect if you were to take a similarly-sized random set of expressed genes?

The authors attempt to present several lines of evidence supporting the biological significance of the network clusters. First they show that genes with somatic mutations in the tumours have greater proximity than would be expected to the cluster's "signature gene", the gene that has greatest centrality in each network cluster. One problem with this is that as noted earlier, the protein clusters contain a small number of large networks, and a large number of small networks. The signature gene is selected from the largest cluster, and it isn't clear from the method how they perform the proximity calculation when the mutated gene falls in one of the smaller networks (which happens frequently, as seen in Supplementary Figures 2-4). Is there simply an ascertainment bias from selecting the largest network, since the chance of a mutated gene falling in a network will be proportional to the number of genes in the network?

The authors then show that the expression level of some set of genes from Cluster 2 are

correlated with overall survival among TCGA patients. (It wasn't clear to me whether this survival signature was derived from the central "signature gene" in Cluster 2, or was derived from all genes in the signature.) In any case, the modest difference in survival observed among the patient groups has multiple confounders and other problems. For one, the difference in survival among the two groups occurs almost immediately, suggesting that perioperative deaths weren't censored from the data set. Second, censored observations aren't indicated on the Kaplan-Meier curve, raising the possibility that these weren't taken into account when calculating the significance. Third, there was no attempt to take into account important clinical covariates such as tumour stage, grade, nodal status or patient age. Fourth, because the survival curve represents multiple tumor types, is there a type-specific confounder? Finally, I am going to guess that most of the tumours used for the Cluster 2 survival signature analysis came from the breast cancer data set. As the authors know, breast cancer is notoriously easy to find prognostic signatures in, and even randomly selected gene signatures are significantly associated with survival (PMID:22028643). So the threshold is not to show that one can derive a survival signature from the cluster, but to show that the signature performs better than multiple randomly selected ones.

The last line of evidence supporting the clinical value of the clustering procedure is that several cell lines responded to drugs that target genes related to the clusters. I had a lot of trouble understanding the significance of these experiments. Three drugs were selected, one targeting a Cluster 2 signature gene, one targeting pathways shared among all three clusters, and one targeting a gene (CHK1/2) not represented in the clusters. The first two drugs showed activity in two cancer-derived cell lines, and the third did not. Also, the first two drugs showed a synergistic activity. The trend is OK, but the sample size (three drugs x two cell lines) is so small that I can't get excited about this. I could imagine that lots of cell lines are sensitive to proteasome and PLK1 inhibitors, and that few are responsive to CHK1/2 inhibitors unless they are already defective in the DNA damage response. In any case, given that all common non-colon tumor types fall into the Cluster 2 group, this result cannot be used to argue that the clustering method has promise for precision medicine.

In short, I am unconvinced of either the novelty or the significance of this work, and feel that it is unsuitable for publication in Nature Communications.

Reviewer #2 (Remarks to the Author):

Faria do Valle et al. present an analysis of expression data in TCGA from 11 different tumor types in combination with protein interaction data. In order to tackle the complexity in the data set, they use a predefined set of cancer-associated genes and cluster tumors based on the co-expression of these genes across samples. They then define three sub-networks in the protein interaction network by weighting the edges using the gene co-expression. These sub-networks are then investigated for their therapeutic potential based on the enrichment of signature genes, drug targets in the neighborhood, discrimination of survival groups. They also provide an experimental investigation of drug sensitivity for 3 compounds on 2 cell lines and their combination.

I think developing novel targeting strategies is a very interesting and relevant challenge in translational cancer genomics and the authors present a neat attempt to combine publicly available gene-expression data and network-based characterization toward identifying potential proteins to be targeted. Overall, the manuscript is clearly written and the methods are explained fairly. That being said, before it becomes available to the broad audience of the journal, I would like to raise several points that are not entirely clear to me and lack additional experiments to strengthen the soundness of the manuscript.

Major points

1. The reproducibility of works is becoming increasingly important in the field of cancer genomics.

Accordingly, it would be extremely useful to demonstrate the robustness of the presented approach. In particular, how do the cancer-related gene selection, underlying protein interaction network and gene co-expression cutoff used in the analysis affect the conclusions?

- a. Would the clusters change substantially if the genes from COSMIC or IntoGen (www.intogen.org) to be used?
 - b. The interactome used in the analysis is based on affinity purification and its coverage can be increased by adding interactions from orthogonal experimental techniques such as Y2H (see Rolland et al., 2014, Cell and Menche et al., 2015, Science).
 - c. How the z cutoff used to define gene co-expression links impact the resulting clusters?
2. The P-values for the statistical enrichment of the functions in Table 4 should be provided (e.g., based on a Fisher's test of number of genes in the cluster & in the pathway). Similarly, are the numbers on Table 5 different from what one would expect from chance? Again the authors should provide the statistical indicators demonstrating the extremeness of the observation on the number of clinical trials with the targets listed on the table in comparison to all the clinical trials in the database.
 3. To strengthen the general applicability of the method and ensure that the authors do not over-fit on existing data, it would definitely help to see whether survival curves are significantly different when a cross-validation approach is used. That is, when half of the patients (224 patients) are used to generate the signatures and the other half is used for k-means clustering. Additionally, the authors can also consider incorporating data from publicly available resources such as CCLE and GDSC to increase the effect sizes, which in turn could help to overcome the observed indifference between survival groups using clusters 1 and 3.
 4. The motivation behind the selection of the drugs for experimental validation is not entirely clear. In particular, among other genes in the cluster 2 signature, why a drug that is targeting PLK1 is chosen? As the authors mention the importance of centrality in the previous section, is it because it had the higher centrality in the network? Can authors come up with a prioritization method based on their analysis that would rank drugs based on their targets or the pathways they are acting through? This is extremely important to clear doubts on potential cherry picking of the three drugs and two cell lines used in the validation.

Minor

1. The introduction can touch base on existing efforts on network-based analysis of cancer genomes, the authors can refer to Creixell et al. 2015, Nat Meth for a range of relevant studies.
2. The authors should explain how the cancer-related genes are curated and mention the publication status for Ontocancro appropriately.
3. Table 2 mentions 591 genes in the network as opposed to 760 genes mentioned earlier, are the remaining 169 genes not in the network?
4. Not clear why the two largest connected components are chosen for cluster 2, while in the other clusters only the largest component is used.
5. I haven't been able to locate how many replicates were used in the sensitivity analysis. Along the same lines, how the confidence interval is decided on Figure 6. It would also be useful to briefly explain how the dose range to be tested are decided for these experiments.
6. Consider adding cluster id as a column to Table 1.
7. Pg3, ln49: factors/regulators choose one. Pg3, ln59: consider removing actually. Pg4, ln72: use the word topological instead of structural to avoid confusion with protein structure based analysis of interactions. Pg7, ln111: "all signatures" is ambiguous (if I understand correctly, they are the ones in the first column of Table 3). Pg7, ln123: across patients. Pg12, ln219: whose expression instead of which expression. Pg15, ln288: *T*umour. Pg17, ln316: "result different" unclear, use differ perhaps.

Reviewer name: Emre Guney

Point-to-point response to referees' comments

Reviewer #1 (Remarks to the Author):

This paper describes a method for combining transcriptomic data with a protein-protein interaction network and a database of cancer-related genes in order to identify gene clusters that are related to the biology of the tumour. The authors demonstrate that their method identifies known cancer related pathways, can be used to find a prognostic biomarker, and that several drugs targeted at members of the clusters have activity against cancer cell lines.

1) Although the subject of the study is one of intense interest to researchers working in the field of cancer genomics, I do not feel that the authors have made the case that their method represents an advance in the field.

There are a very large number of network-based methods that have previously been applied to cancer (PMID:22962493, PMID:20529912, PMID:22174262, PMID:23228031, PMID:28388297 (review), PMID:26125594 (review), PMID:28122019, PMID:18990722, PMID:21576238, PMID:20169195).

Most are able to identify enriched cancer-related pathways, select central genes, identify tumour subtypes and (often) identify prognostic biomarkers. The authors have not attempted to differentiate their algorithm from these earlier methods (and the many others; see the reviews for a list!)

We thank the reviewer for presenting a list of state-of-the art papers on the topics of network-based approaches. We agree that our work is clearly embedded under the hat of "network-based methods" and is surely not the first one nor the last on this field.

Going into details of the mentioned papers, anyway we notice that 1) all of them have been applied only to specific case studies (one or two tumour types) and none of them integrates the information coming from multiple tumour types, but can be applied only to one tumour type at a time; 2) most of these approaches propose gene clusters and pathways as biomarkers, differing from our model in which single nodes are identified as a part of a signature derived from a combination of topological properties of the network and experimental data; 3) most of the proposed papers (and many others) can be thought as "seed-based" models (like HotNet, Vandin et al J. Comp. Biol. 18(3) 2011) in which an initial set of significant genes or modules must be provided to proceed with a network-based analysis, an approach completely different from ours in which no a priori modules or pathways need to be set as input, but they emerge spontaneously from the analysis pipeline.

We remark that our objective is drug repurposing among tumours, so it is necessary to devise a procedure to combine information from multiple tumours, rather than just performing a network analysis on a single tumour, and in this field of multi-tumour analysis the examples are up to now much fewer.

We have added the suggested references in the first part of the paper, with detailed comments to highlight that our approach is part of a class of algorithms (as we also described in an our recent review paper Bersanelli et al. BMC Bioinformatics 2016, 17(Suppl 2):15) but addressing a different aim.

2) Even on their own terms, the results presented in this paper are not overwhelming. A non-novel hierarchical clustering process (Figure 1), produces the three tumour type clusters that are examined in the rest of the paper.

These clusters are very unbalanced. 85% of the tumours were placed in Cluster 2, 10% are in Cluster 1, and just 5% are in Cluster 3. Given that the vast majority of tumour types are in cluster 2, the method is not providing a finely detailed map of tumour type relatedness.

Apparently all tumours of the same type clustered together, and the main surprise at this step of the analysis is that brain low grade glioma co-clustered with the kidney tumours rather than with glioblastoma multiforme. I would have been interested in some discussion of this finding.

The reviewer is commenting on a not novel clustering approach and on a possible lack of (maybe biological) relatedness between the clustered tumours.

Even if the specific clustering technique (hierarchical clustering) is not novel, it is the whole analysis pipeline (from clustering to network-based signature extraction) that provides an original approach to multi-tumour analysis. Moreover, for our clustering we are considering the whole correlation matrices of sample expression profiles for each tumour, and not single-sample expression profile vectors, and this is not so common in many clustering approaches. Moreover, even if it might not be completely original, our clustering analysis successfully combines together tumour types for a drug repurposing scope (which is actually the main aim of our paper).

The unbalance of cluster sizes (2/11 18%, 6/11 54%, and 3/11 27% for cluster 1, 2, 3 respectively) can not be considered as a negative result of our analysis, because it is not under our control (our clustering is completely unsupervised) and might be simply due to the fact that only a small sample of all possible tumour types is considered in our analysis, thus their number and size could change when a more complete tumour database is used.

Also the biological relatedness of the tumours is not a crucial issue, since our purpose is not to obtain a new tumour biological classification, but we are simply searching for functional relationships between them that can allow us to repurpose drugs. Other papers have shown (es. Ciriello et al, Nature Genetics 45(10) 2013, ref. 3 in our paper; Hoadley et al. Cell 2013 ref. 2; Martinez et al, ref. 18) that a functional classification might combine together tumours of different types, or might separate tumours that should be biologically or anatomically more similar by common clinical approaches.

Regarding tumour clustering, Hoadley et al, 2013 (Cell) performed a multiplatform analysis of 12 cancer types. They did not study low-grade glioma LGG, but by clustering samples according to the gene expression and copy number variation profiles, they also observed that kidney renal papillary KIRP and glioblastoma multiforme GBM clustered in the same group. We added a comment in the Discussion section.

3) When these transcription-based clusters are superimposed on a protein-protein interaction network it results in...three protein network clusters. Pathway

overrepresentation analysis reveals three overrepresented well-known cancer-related biological processes (a non-novel finding), but the genes representing the processes are distributed relatively evenly among the three network clusters. When you examine the details of the protein-based network clusters, it emerges that there are many networks within each cluster: something on the order of 50 networks in cluster 2 (eyeballing Supplementary Figure 3). It is unclear to me what value this additional step of protein-based network clustering is adding to our understanding of the transcription-based tumour type clusters? At the least, I'd like to see some correlative data that shows that these networks have biological relevance; for example, when you select the protein-based network clusters based on the transcriptional profiles, are their nodes more connected than you would expect if you were to take a similarly-sized random set of expressed genes?

Our approach superimposed on a protein interaction network (PPI, common to all considered tumours) three cluster-specific correlation matrices, with samples grouped by hierarchical clustering procedure described before. With this processing, we obtained three networks (combination of PPI topological structure and gene expression experimental data) with nodes and links specific to each tumour cluster, onto which to measure node centralities. This allowed us to extract three cluster-specific gene signatures as top-ranking nodes (the first decile of the gene lists ranked by node Spectral Centrality).

At a biological pathway level, our analysis highlighted some biological processes which are very relevant in cancer (and thus already known as noted by the reviewer) but our main original result is that we identified specific gene signatures as possible drug targets common to several types of cancers, thus allowing drug repurposing. These signatures are found through a network centrality measure, that requires a network structure to be defined: the protein-based network clustering described above is the preliminary processing to obtain the signatures.

Regarding the role of the protein-based network clustering step, we have shown that the PPI network alone (i.e. without overlapping cluster-specific gene expression correlation matrices) produces an unique gene signature (thus common to all tumours) with small overlap with the cluster-specific gene signatures (3/25, 13/27 and 4/24 common genes between the signature of PPI network alone and those of clusters 1, 2, 3, respectively, as described in the main text), thus the combination of PPI network topology with experimental measures of gene expression profiling is necessary to exploit the cancer-specific information embedded in the network.

Regarding node connectivity, we remark that our approach corresponds to a "percolation" phenomenon on the original protein network (removing non significant links due to negative CLR z-score) thus the number of links is necessarily reduced, and the node connectivity correspondingly, and this would happen also for randomly chosen signature genes.

Since the biological meaning of our signature genes is reflected in their connectivity patterns, as an example we have shown in the Supplementary Section "Effect of gene expression correlation on PPI profiles" how BTRC gene (which is in the signatures of cluster 1 and 2) has different neighbours in the two clusters.

4) The authors attempt to present several lines of evidence supporting the biological significance of the network clusters. First they show that genes with somatic mutations in the tumours have greater proximity than would be expected to the cluster's "signature gene", the gene that has greatest centrality in each network cluster. One problem with this is that as noted earlier, the protein clusters contain a small number of large networks, and a large number of small networks. The signature gene is selected from the largest cluster, and it isn't clear from the method how they perform the proximity calculation when the mutated gene falls in one of the smaller networks (which happens frequently, as seen in Supplementary Figures 2-4). Is there simply an ascertainment bias from selecting the largest network, since the chance of a mutated gene falling in a network will be proportional to the number of genes in the network?

In response to referee's question, we remark that for each cluster we obtain (by our approach) a list of signature genes corresponding to the first decile of the centrality measures (25, 27, and 24 genes for clusters 1, 2, 3, respectively, Table 3 of the main text) performed on the largest network connected components, and not a single signature gene. Probably there was a misspelling that caused this misunderstanding: we have restated the sentences in a clearer manner.

In each cluster we calculated the average distance of all signature genes with each mutated gene in the network. Since most (if not all) network-based approaches, when dealing with a network composed of several disconnected components, typically apply the analyses only to the largest cluster in the network (Giant Component or the largest components if they are similar in size), discarding the smallest subnetworks (e.g. in the paper by Goh et al regarding disease network, PNAS May 22, 2007 vol. 104 no. 21 8685–8690) we applied the same procedure accordingly, thus all of our calculation refer to the largest components only. In our opinion, the role of network approaches is to highlight relevant elements inside large systems, since in small systems the choice can be made "by hand" or simply by testing all the element properties.

Anyway, as the referee notes, there are small clusters (composed by 2-3 nodes) that show the relevant features we are looking for, namely to have a mutated gene as a part of the component, thus possibly also these small clusters could contain relevant targets for drug repurposing. We have added additional supplementary tables 9-11 (as sheets of Supplementary Excel file) containing centrality measures also for genes not contained in the cluster giant components, but belonging to components with at least one mutated gene, so that the readers can have information also on possible relevant genes found in the smaller network components.

Moreover, in the supplementary Material ("Distance between signature genes and mutated genes") we verified that the giant components were not significantly enriched in mutated genes in comparison to the whole network, thus considering the giant components only did not introduce a bias in this sense.

5) The authors then show that the expression level of some set of genes from Cluster 2 are correlated with overall survival among TCGA patients. (It wasn't clear to me whether this survival signature was derived from the central "signature gene" in Cluster 2, or was derived from all genes in the signature.) In any case, the modest difference in survival observed among the patient groups has multiple confounders

and other problems. For one, the difference in survival among the two groups occurs almost immediately, suggesting that perioperative deaths weren't censored from the data set. Second, censored observations aren't indicated on the Kaplan-Meier curve, raising the possibility that these weren't taken into account when calculating the significance.

Third, there was no attempt to take into account important clinical covariates such as tumour stage, grade, nodal status or patient age.

Fourth, because the survival curve represents multiple tumor types, is there a type-specific confounder? Finally, I am going to guess that most of the tumours used for the Cluster 2 survival signature analysis came from the breast cancer data set. As the authors know, breast cancer is notoriously easy to find prognostic signatures in, and even randomly selected gene signatures are significantly associated with survival (PMID:22028643).

So the threshold is not to show that one can derive a survival signature from the cluster, but to show that the signature performs better than multiple randomly selected ones.

As suggested by the referee, we refined our survival analysis by considering data censoring, and modified Figure 5 in the main text accordingly (even if we decided not to plot the censored data for better picture clarity), and we better clarified in the text that the whole gene signature of cluster 2 (composed of 27 genes) was used for sample clustering and survival analysis. When we included data censoring, the significance of our analysis increased (from $P = 4.54 \times 10^{-3}$ to $P = 7.26 \times 10^{-18}$). In the new figure 5 it is shown how the tumour samples are distributed 1) per tumour type and 2) inside each cluster. As it can be seen, the breast cancer samples do not represent an overwhelmingly fraction of sample set, being less than 25%, with a similar fraction of ovarian cancer OV and glioblastoma GBM samples which should not suffer from the same bias.

Moreover, we performed additional analyses as suggested by the referee 1) estimating the role of possible covariates such as age and tumour type, and 2) comparing our signatures with random signatures (described in a section of Supplementary Material called "Survival Analysis").

A new analysis was performed with 500 random signatures, and the results are given in a 2-d plot combining the statistical significance of each single survival analysis (P -value) with an index - S_T - describing the distribution of tumour samples between the two classes. This analysis shows that the clusters obtained from the original signature contain balanced fractions of samples from all tumours, and are not strongly biased towards containing only some specific tumour samples in one class and other tumour types in the other. As it can be seen, there are signatures that perform better than ours, that anyway result more biased in terms of segregation of specific tumours in only one class, thus characterized by the type-specific confounder bias suggested by the reviewer. Our signature shows a good balance between survival stratification performance and balance between tumour types inside the two classes, and it is among the top-ranking signatures with sufficiently balanced classes ($ST < 0.5$, 95th percentile, thus in the top 5% performing signatures).

Moreover, in the supplementary section we show that the possible confounders (age and tumour type) may contribute to the significance of the survival curve, but to a lesser extent than our signature-based clustering.

6) The last line of evidence supporting the clinical value of the clustering procedure is that several cell lines responded to drugs that target genes related to the clusters. I had a lot of trouble understanding the significance of these experiments. Three drugs were selected, one targeting a Cluster 2 signature gene, one targeting pathways shared among all three clusters, and one targeting a gene (CHK1/2) not represented in the clusters. The first two drugs showed activity in two cancer-derived cell lines, and the third did not. Also, the first two drugs showed a synergistic activity. The trend is OK, but the sample size (three drugs x two cell lines) is so small that I can't get excited about this. I could imagine that lots of cell lines are sensitive to proteasome and PLK1 inhibitors, and that few are responsive to CHK1/2 inhibitors unless they are already defective in the DNA damage response.

In any case, given that all common non-colon tumor types fall into the Cluster 2 group, this result cannot be used to argue that the clustering method has promise for precision medicine.

We agree that our validation set is far from being exhaustive, but we want to remark strongly that no experimental validation of single or multi-tumour signatures identified by systems biology approaches has been previously reported in the papers brought as a comparison with ours.

*The cellular models used in this work were selected as follows: a very common tumour (breast cancer cell line MCF-7) and a very aggressive one (glioblastoma cell line TG98). Regarding the target genes, our choice was based on the list provided by our gene signatures, considering all genes ranking in the first decile as almost equally important, since ranking could be affected by uncertainties in the exact structure of PPI network, in the value of expression genes, and by other experimental factors. This list was further filtered based on the cell culture conditions required for validation experiments. We chose drugs allowing straightforward testing in terms of experimental conditions. For example, the first ranking gene (IL6R) should have needed a complex experimental setting taking into account also the microenvironment (e.g. co-culturing with stromal cells and adding cytokines to the medium). Designing novel drugs for the newly proposed targets is far beyond the scopes of our studies, so we had to find a compromise between top-ranking genes (according to our signatures) and possibility to have clear and informative results. Therefore we chose PLK1 and proteasome inhibitors. We agree with the reviewer that a number of tumor models will be sensitive to proteasome inhibitors. Indeed, genes belonging to the ubiquitin proteasome system can be found in all clusters. A number of cell lines may also respond to PLK1 inhibition. The sensitivity of the selected models might also be related to the expression levels of PLK1, which is higher in cluster 2 tumors compared to the others. By literature interrogation, we identified many cell lines which are insensitive to Bortezomib (Lü S et al. *Exp Hematol.* 2009; Fuchs D et al. *J Cell Biochem.* 2008; Oerlemans R et al. *Blood.* 2008; Lü S et al. *J Pharmacol Exp Ther.* 2008; Rückrich T et al. *Leukemia.* 2009; Balsas P et al. *Leuk Res.* 2012; Ri M et al. *Leukemia.* 2010; Pérez-Galán P et al. *Blood.* 2011; Franke NE et al. *Nature* 2012; de Wilt LHAM et al. *Biochem Pharmacol.* 2012; Wu Y-X et al. *Oncotarget.* 2016). Moreover, studies suggested that the sensitivity to proteasome inhibitor is associated to the drug nature (e.g., the same cell B16F10 has been highly sensitive to Bortezomib [Yerikaya A et al. *Mol Med Rep.* 2010] and highly resistant to quercetin [Rodriguez J et al. *Melanoma Res.* 2002]). In parallel, diverse cells are insensitive to PLK1 inhibitor, and Nonomiya et al 2016 describe some resistance mechanisms exploring the possibility to identify predictable*

biomarkers of PLK inhibitors (Nonomiya Y et al. *Cancer Sci.* 2016). Regarding CHK1 inhibitor, we agree with the reviewer that CHK1 inhibitor is largely used in combination therapies due to its function as chemo- and radiosensitizer. In parallel, oncogenes causing replication stress, along with defects in nucleotide excision repair, non-homologous end-joining and homologous recombination DNA repair sensitize cells to CHK1 inhibition, and this inhibitor is used as single agent in ongoing clinical trials as reviewed by Rundle et al. (Rundle S. et al, *Cancers.* 2017). Since this drug was available in our lab, we decided to use it as a negative control: one drug is not a sufficiently large statistical sample, but anyway the direction of our results was correct both for the positive and negative cases. Moreover, even the observed synergistic effects on both cell lines (a non-trivial result) suggest that our signatures can provide hints also for novel drug combinations. We hope to have expanded enough our comments, that have consequently been added to the main text. Taken together, our *in vitro* results offer a proof of concept supporting our network-based analyses, and in our opinion the results obtained in our models is highly encouraging.

Regarding the last referee comment, the possibility to extend our results towards personalized medicine at the moment is not supported by experimental evidence, since our analysis is valid for tumour types and not for single samples. There is a sentence in the end of the discussion pointing to possible future research in this direction, but if the referee and the editors agree that is too strong a statement, we can remove it from the final conclusion since it is purely speculative and only points towards possible future achievements.

References

- Balsas P, Galán-Malo P, Marzo I, Naval J. Bortezomib resistance in a myeloma cell line is associated to PSM β 5 overexpression and polyploidy. *Leuk Res.* 2012;36: 212–218. doi:10.1016/j.leukres.2011.09.011
- de Wilt LHAM, Jansen G, Assaraf YG, van Meerloo J, Cloos J, Schimmer AD, et al. Proteasome-based mechanisms of intrinsic and acquired bortezomib resistance in non-small cell lung cancer. *Biochem Pharmacol.* 2012;83: 207–217. doi:10.1016/j.bcp.2011.10.009
- Franke NE, Niewerth D, Assaraf YG, van Meerloo J, Vojtekova K, van Zantwijk CH, et al. Impaired bortezomib binding to mutant β 5 subunit of the proteasome is the underlying basis for bortezomib resistance in leukemia cells. *Leukemia.* 2012;26: 757–768. doi:10.1038/leu.2011.256
- Fuchs D, Berges C, Opelz G, Daniel V, Naujokat C. Increased expression and altered subunit composition of proteasomes induced by continuous proteasome inhibition establish apoptosis resistance and hyperproliferation of Burkitt lymphoma cells. *J Cell Biochem.* 2008;103: 270–283. doi:10.1002/jcb.21405
- Lü S, Yang J, Song X, Gong S, Zhou H, Guo L, et al. Point Mutation of the Proteasome β 5 Subunit Gene Is an Important Mechanism of Bortezomib Resistance in Bortezomib-Selected Variants of Jurkat T Cell Lymphoblastic Lymphoma/Leukemia Line. *J Pharmacol Exp Ther.* 2008;326. Available: <http://jpet.aspetjournals.org/content/326/2/423.short>
- Lü S, Yang J, Chen Z, Gong S, Zhou H, Xu X, et al. Different mutants of PSMB5 confer varying bortezomib resistance in T lymphoblastic lymphoma/leukemia cells derived from the Jurkat cell line. *Exp Hematol.* 2009;37: 831–837. doi:10.1016/j.exphem.2009.04.001
- Nonomiya Y, Noguchi K, Tanaka N, Kasagaki T, Katayama K, Sugimoto Y. Effect of AKT3 expression on MYC- and caspase-8- dependent apoptosis caused by polo- like kinase inhibitors in HCT 116 cells. *Cancer Sci.* 2016;107: 1877–1887. doi:10.1111/cas.13093
- Oerlemans R, Franke NE, Assaraf YG, Cloos J, van Zantwijk I, Berkers CR, et al. Molecular basis of bortezomib resistance: proteasome subunit β 5 (PSMB5) gene mutation and overexpression of PSMB5 protein. *Blood.* 2008;112.
- Pérez-Galán P, Mora-Jensen H, Weniger MA, Shaffer AL, Rizzatti EG, Chapman CM, et al. Bortezomib resistance in mantle cell lymphoma is associated with plasmacytic differentiation. *Blood.* 2011;117.

- Ri M, Iida S, Nakashima T, Miyazaki H, Mori F, Ito A, et al. Bortezomib-resistant myeloma cell lines: a role for mutated PSMB5 in preventing the accumulation of unfolded proteins and fatal ER stress. *Leukemia*. 2010;24: 1506–1512. doi:10.1038/leu.2010.137
- Rodriguez J, Yáñez J, Vicente V, Alcaraz M, Benavente-García O, Castillo J, et al. Effects of several flavonoids on the growth of B16F10 and SK-MEL-1 melanoma cell lines: relationship between structure and activity. *Melanoma Res*. 2002;12: 99–107.
- Rückrich T, Kraus M, Gogel J, Beck A, Ovaa H, Verdoes M, et al. Characterization of the ubiquitin–proteasome system in bortezomib-adapted cells. *Leukemia*. 2009;23: 1098–1105. doi:10.1038/leu.2009.8
- Rundle S, Bradbury A, Drew Y, Curtin NJ. Targeting the ATR-CHK1 Axis in Cancer Therapy. *Cancers* 2017, 9(5), 41; doi:10.3390/cancers9050041
- Wu Y-X, Yang J-H, Saitsu H, Wu Y-X, Yang J-H, Saitsu H. Bortezomib-resistance is associated with increased levels of proteasome subunits and apoptosis-avoidance. *Oncotarget. Impact Journals*; 2016;7: 77622–77634. doi:10.18632/oncotarget.12731
- Yerlikaya A, Okur E, Semih Şeker, Erin N. Combined effects of the proteasome inhibitor bortezomib and Hsp70 inhibitors on the B16F10 melanoma cell line. *Mol Med Rep*. 2010;3: 333–339. doi:10.3892/mmr

Reviewer #2 (Remarks to the Author):

Faria do Valle et al. present an analysis of expression data in TCGA from 11 different tumor types in combination with protein interaction data. In order to tackle the complexity in the data set, they use a predefined set of cancer-associated genes and cluster tumors based on the co-expression of these genes across samples. They then define three sub-networks in the protein interaction network by weighting the edges using the gene co-expression. These sub-networks are then investigated for their therapeutic potential based on the enrichment of signature genes, drug targets in the neighborhood, discrimination of survival groups. They also provide an experimental investigation of drug sensitivity for 3 compounds on 2 cell lines and their combination.

I think developing novel targeting strategies is a very interesting and relevant challenge in translational cancer genomics and the authors present a neat attempt to combine publicly available gene-expression data and network-based characterization toward identifying potential proteins to be targeted. Overall, the manuscript is clearly written and the methods are explained fairly. That being said, before it becomes available to the broad audience of the journal, I would like to raise several points that are not entirely clear to me and lack additional experiments to strengthen the soundness of the manuscript.

Major points

1. The reproducibility of works is becoming increasingly important in the field of cancer genomics. Accordingly, it would be extremely useful to demonstrate the robustness of the presented approach.

In particular, how do the cancer-related gene selection, underlying protein interaction network and gene co-expression cutoff used in the analysis affect the conclusions?

a. Would the clusters change substantially if the genes from COSMIC or IntoGen (www.intogen.org) to be used?

We repeated the tumor clustering procedure using the expression profiles of the High Confidence Driver genes in the IntoGen list TCGA pan-cancer12. The clusters are different (shown in the figure below, only for the reviewer) even if they partially agreed with our results, since we also observed the following groups of tumors clustered together: COAD and READ; LUAD, BRCA, LUSC, GBM, and OV. We remark that COSMIC and IntoGen provides somatic mutational data, while our clustering was based on tumor transcriptional profiling. As these two data types might represent different underlying oncogenic processes, it is expected that the clustering results might differ when changing from one type of data to another.

b. The interactome used in the analysis is based on affinity purification and its coverage can be increased by adding interactions from orthogonal experimental techniques such as Y2H (see Rolland et al., 2014, Cell and Menche et al., 2015, Science).

Following referee's suggestion, we extended our analysis to a PPI obtained from our original Bioplex-Ontocancro network combined with the PPI networks described in the mentioned papers. Results are fully described in an additional Supplementary section ("Impact of PPI network on gene signatures") and discussed in the main text: the new signatures are in good agreement with the previous, with >50% overlap of signature genes for all clusters. In particular, the signature gene in cluster 2 we chose for drug targeting is also in the signature of the extended PPI.

c. How the z cutoff used to define gene co-expression links impact the resulting clusters?

We expanded our analysis applying a set of increasing z-score thresholds, and the resulting clusters showed the same overall pattern, thus resulting quite robust to a change in z-score cutoff. We added a section in the supplementary material ("Impact of CLR z-score thresholds on tumour clustering") showing the results of clustering for different z cutoff threshold values, that also describes how link density decreases as a function of cutoff, and thus how much the original topology is affected by such thresholding. For a large interval of z cutoff values, the clusters are in good agreement with the initial ones, supporting the robustness of our results.

2. The P-values for the statistical enrichment of the functions in Table 4 should be provided (e.g., based on a Fisher's test of number of genes in the cluster & in the pathway). Similarly, are the numbers on Table 5 different from what one would expect from chance? Again the authors should provide the statistical indicators demonstrating the extremeness of the observation on the number of clinical trials with the targets listed on the table in comparison to all the clinical trials in the database.

Regarding Table 4, the pathways were manually curated, by considering gene information from Uniprot and GeneCards databases. We have added this explanation to the main text, and we also included Supplementary tables 6-8 with an enrichment analysis performed on Gene Ontology, showing that the top-ranking pathways are the same.

Regarding the results in Table 5, it is hard to perform an enrichment analysis of genes annotated in ClinicalTrials.org, since the information on the signature genes could not be obtained by automatic query of the database (that is actually not structured for such analyses), thus it would be infeasible to manually recover such information for the hundreds of genes in our full network.

3. To strengthen the general applicability of the method and ensure that the authors do not over-fit on existing data, it would definitely help to see whether survival curves are significantly different when a cross-validation approach is used. That is, when half of the patients (224 patients) are used to generate the signatures and the other half is used for k-means clustering. Additionally, the authors can also consider incorporating data from publicly available resources such as CCLE and GDSC to increase the effect sizes, which in turn could help to overcome the observed indifference between survival groups using clusters 1 and 3.

We have extended our analysis of survival curves, merging these requests with those of another reviewer asking a more detailed description of the data used and the possible confounding factors: 1) we added data censoring to survival analysis; 2) we have considered the role of other cofactors where available (such as age and

cancer type); 3) as a procedure to test signature robustness, we have compared the significance of cluster 2 signature with randomly generated ones, considering both P -value and a newly introduced index - S_T - described in the Supplementary Material, that estimated the possible imbalance of tumour samples in the two classes that might affect performance. In this context, we can see that 1) data censoring increases p -value significance of the real signature (from $P = 4.54 \times 10^{-3}$ to $P = 7.26 \times 10^{-18}$); 2) our signature is one of the best performing (in the top 5%) among the random signatures with a comparable value of ST (<0.5), thus answering to referee's request about signature crossvalidation; 3) the confounding factors have an effect, that is anyway smaller than our clustering based on the signature. All the results of these analyses have been added to a Section of the Supplementary Material ("Survival Analysis"), and also Figure 5 in the text has been changed according to the new analysis (considering censored data, even if not shown for picture clarity, and showing the ratio of tumour samples found in both sample groups).

4. The motivation behind the selection of the drugs for experimental validation is not entirely clear. In particular, among other genes in the cluster 2 signature, why a drug that is targeting PLK1 is chosen? As the authors mention the importance of centrality in the previous section, is it because it had the higher centrality in the network? Can authors come up with a prioritization method based on their analysis that would rank drugs based on their targets or the pathways they are acting through? This is extremely important to clear doubts on potential cherry picking of the three drugs and two cell lines used in the validation.

In principle, our best prioritization list would correspond to the list of genes ranked by spectral centrality, since this is the index we use in our analysis (we have shown all genes centrality values in Supplementary Tables 9-11). In our opinion, it must be taken into account that some "noise" can be present in the ranking due to wrong/unknown PPI links, error in gene expression level measurements, limited/unbalanced number of samples, and other factors, so some sort of "blurring" must be expected in our sorted list of target genes: our choice of the first decile of centralities could be a reasonable threshold to define almost equivalent drug targets, even if a more refined choice could be made with added biological and clinical knowledge, as we in fact did. As an example, the analysis with the extended PPI (as described in response to question 1b) produced signatures with an overlap of about 50% genes, and our chosen target PLK1 was found in both, even if it was not the top-ranking gene. We could verify this a posteriori, but it confirms our choice not to rely necessarily on exact gene ranking by centrality, but also on additional biological information.

In particular, our selection on target to be validated was combined with many other occurring factors, the most important of which is the existence of drugs for a specific target, since new drug design is far beyond the scope of our paper. This list was further filtered based on cell culture conditions required for feasibility of validation experiments in our laboratories. We chose drugs allowing straightforward testing in terms of experimental conditions. For example, the first ranking gene (IL6R) should have needed a complex experimental setting taking into account also the microenvironment (e.g. co-culturing with stromal cells, and addition of cytokines to the medium). The non-targeting drug was available in our lab, so we decided to use it

as a negative control, even if we are aware that one drug is not a sufficiently large statistical sample. In summary, the direction of our results was correct both for the positive and negative cases, and in particular we remark the positive results in terms of drug synergy for both cell lines, that make us confident that our network-based signature could be helpful for target gene prioritization also for multi-drug therapies. We are very aware that our choice of validation cells and drugs is far from being exhaustive, but it can anyway constitute a "proof of concept" not so commonly found in many other network-based analyses (for example all the papers we have referenced in the introduction with similar purposes do not show any kind of in vitro validation, neither in single-tumour studies).

Minor

1. The introduction can touch base on existing efforts on network-based analysis of cancer genomes, the authors can refer to Creixell et al. 2015, Nat Meth for a range of relevant studies.

We have added a paragraph in the introduction section mentioning the literature state of art for network approaches in cancer data analysis (comprising the suggested reference). We also increased our list of references of other network-based approaches

2. The authors should explain how the cancer-related genes are curated and mention the publication status for Ontocancro appropriately.

We added the reference as requested: Librelotto, G. R. ; Mombach, J. C. M. ; Sinigaglia, M. ; Simão, E. ; Cabral, H. B. ; Castro, M. A. A. ; An Ontology to Integrate Transcriptomics and Interactomics Data involved in Gene Pathways of Genome Stability. BSB 2009, LNBI 5676, pp. 164-167, 2009.

3. Table 2 mentions 591 genes in the network as opposed to 760 genes mentioned earlier, are the remaining 169 genes not in the network?

760 genes is the number of genes annotated both in Bioplex and Ontocancro: this was the list of gene used to construct the correlation matrices for clustering tumours. When mapped onto the Bioplex network structure, 169 genes were isolated nodes, so they were removed from the following network analyses, resulting in a network with 591 nodes (divided between giant and smaller connected components). We have specified it more clearly in the main text, Methods Section, and in the Supplementary Section "Tumour cluster networks description".

4. Not clear why the two largest connected components are chosen for cluster 2, while in the other clusters only the largest component is used.

We considered the two largest connected components for two reasons: 1) in order to have approximately the same number of nodes for each cluster, and 2) because in this case the two largest components had almost the same size (much larger than the other components) an unusual situation as compared to typical Random-Network null model of percolation, in which a unique giant component emerges. We have explained this choice in the main text accordingly.

5. I haven't been able to locate how many replicates were used in the sensitivity analysis. Along the same lines, how the confidence interval is decided on Figure 6. It would also be useful to briefly explain how the dose range to be tested are decided for these experiments.

Each experiment in the sensitivity analysis was done in triplicates. The dose range was chosen based on insights from the literature on the same cell lines or related cellular models (Bhola NE et al. Cancer Research 2015; Thaler S et al. IJC 2014; Pezuk JA et al. Cancer Biotherapies and Radiopharmaceuticals 2013; Vlachostergios PJ et al. Cell Mol Neurobiol 2013; Hideshima H et al. International J of Oncology 2014; Yao F et al. Mol Medicine Reports 2012; Han J et al. Breast Cancer Res 2009; Tang Y. et al. Cancer Biology & Therapy 2012; Bryant C et al. BMC Cancer 2014; Ma Z et al. Mol Medicine Reports 2012) and then adjusted in order to allow tracking of the cell viability curve and definition of the IC50. The IC50 was calculated using Graphpad prism with a 95% confidence interval. These informations have been fully included in the text of the Methods Section.

6. Consider adding cluster id as a column to Table 1.

Done

7. Pg3, ln49: factors/regulators choose one. Pg3, ln59: consider removing actually. Pg4, ln72: use the word topological instead of structural to avoid confusion with protein structure based analysis of interactions. Pg7, ln111: "all signatures" is ambiguous (if I understand correctly, they are the ones in the first column of Table 3). Pg7, ln123: across patients. Pg12, ln219: whose expression instead of which expression. Pg15, ln288: *T*umour. Pg17, ln316: "result different" unclear, use differ perhaps.

Done

References

- Bhola NE, Jansen VM, Bafna S, Giltane JM, Balko JM, Estrada MV et al. Kinome-wide Functional Screen Identifies Role of PLK1 in Hormone-Independent, ER-Positive Breast Cancer. Cancer Research 2015; Jan 15;75(2):405-14. doi: 10.1158/0008-5472.CAN-14-2475.*
- Bryant C, Rawlinson R, Massey AJ. Chk1 Inhibition as a novel therapeutic strategy for treating triple-negative breast and ovarian cancers. BMC Cancer 2014;Aug 7;14:570. doi: 10.1186/1471-2407-14-570.*
- Han J, Ma I, Hendzel MJ, Allalunis-Turner J. The cytotoxicity of γ -secretase inhibitor I to breast cancer cells is mediated by proteasome inhibition, not by γ -secretase inhibition. Breast Cancer Res 2009;11(4):R57. doi: 10.1186/bcr2347.*

- Hideshima H, Yoshida Y, Ikeda H, Hide M, Iwasaki A, Anderson KC, Hideshima T. IKK β inhibitor in combination with bortezomib induces cytotoxicity in breast cancer cells. *Int J Oncol.* 2014 Apr;44(4):1171-6. doi: 10.3892/ijo.2014.2273.
- Ma Z, Yao G, Zhou B, Fan Y, Gao S, Feng X. The Chk1 inhibitor AZD7762 sensitises p53 mutant breast cancer cells to radiation in vitro and in vivo. *Mol Medicine Reports* 2012;Oct;6(4):897-903. doi: 10.3892/mmr.2012.999.
- Pezuk JA, Brassesco MS, Morales AG, de Oliveira JC, de Oliveira HF, Scrideli CA, Tone LG. Inhibition of polo-like kinase 1 induces cell cycle arrest and sensitizes glioblastoma cells to ionizing radiation. *Cancer Biotherapies and Radiopharmaceuticals* 2013;Sep;28(7):516-22. doi: 10.1089/cbr.2012.1415.
- Thaler S, Thiede G, Hengstler JG, Schad A, Schmidt M, Sleeman JP. The proteasome inhibitor Bortezomib (Velcade) as potential inhibitor of estrogen receptor-positive breast cancer. *Int J Cancer.* 2015 Aug 1;137(3):686-97. doi: 10.1002/ijc.29404.
- Tang Y, Dai Y, Grant S, Dent P. Enhancing CHK1 inhibitor lethality in glioblastoma. *Cancer Biology & Therapy* 2012; Apr;13(6):379-88. doi: 10.4161/cbt.19240.
- Vlachostergios PJ, Hatzidaki E, Stathakis NE, Koukoulis GK, Papandreou CN. Bortezomib downregulates MGMT expression in T98G glioblastoma cells. *Cell Mol Neurobiol.* 2013 Apr;33(3):313-8. doi: 10.1007/s10571-013-9910-2.
- Yao F, Wang G, Wei W, Tu Y, Tong H, Sun S. An autophagy inhibitor enhances the inhibition of cell proliferation induced by a proteasome inhibitor in MCF-7 cells. *Mol Medicine Reports* 2012;Jan;5(1):84-8. doi: 10.3892/mmr.2011.590.

Reviewers' comments:

Reviewer #1 (Remarks to the Author):

I very much appreciate both the chance to review this revised paper, the authors' detailed response to the previous reviews, and the substantive changes that the authors made to the manuscript. I continue to think that this paper addresses a significant challenge in cancer genomics, and I feel that the revisions have cleared up several of the communications problems that contributed to my unfavourable previous review.

Nevertheless, I'm afraid I remain unconvinced of the main assertion in the paper that the method of network clustering and subsequent network analysis that the authors describe contributes significantly to our understanding of clinical behaviour. This assertion rests on two key experiments: the discovery that the expression level of genes comprising the Cluster 2 signature are prognostic, and the sensitivity of two cell lines derived from tumour types in Cluster 2 to a pair of drugs that target Cluster 2 signature genes or processes.

Compared to the original version, the prognostic study of survival is better explained and more strongly supported. However, the right hand panel of Figure 5 discloses a serious confounding issue. Among the tumour types that account for the majority of the tumours analysed, the "bad prognosis" signature is present in roughly 80% of GBM cases, in 40% of BRCA cases, and about 20% of OV cases. The 5-year survival time for GBM is dismal, about 10%, whereas breast cancer and ovarian cancer have much better prognoses. The authors made a good start with the multivariate analysis of prognostic factors (Supplementary Table 5) which showed that the signature remained significant even after regressing out the tumour type, but they may have missed the fact that within the TCGA data set for each tumour type there are differences in tumour stage and grade, which are strong clinical prognostic factors. To make a convincing case, the authors need to perform the survival analysis on each tumour type separately, and to perform a covariate analysis that at least takes into account tumour stage.

Compared to the original version, the drug sensitivity experiment has not significantly changed. I still find this study of two cell lines across three drugs to be woefully inadequate to support the striking claim that the network analysis method identifies novel drug targeting strategies. The problem is that there is no negative control. As a reader, I cannot tell what is the expected rate of success for any two randomly chosen cell lines and three randomly selected targeted small molecule agents. To do a convincing validation experiment without actually designing a novel therapeutic (which I agree is outside the scope of this paper!), I would suggest that the authors go to one of the several published drug sensitivity databases in which a variety of cell lines have been systematically screened with large number of targeted and conventional agents, and show that the observed rate of positive drug sensitivity results involving predicted pairs of targeted agents and cell lines derived from particular tumour types exceeds what would be predicted by chance. There are several such databases that spring to mind, including the well-studied NCI-60 cell lines, the Sanger Institute's Genomics of Drug Sensitivity in Cancer project (<http://www.cancerrxgene.org/>), and the Broad Institute's Cancer Therapeutics Response Portal (<https://portals.broadinstitute.org/ctrp/>)

In conclusion, while the manuscript is greatly improved, two key claims remain deficient, and more work needs to be done before I would find this paper acceptable for publication.

Reviewer #2 (Remarks to the Author):

I thank the authors for their effort on addressing my concerns. In my opinion, there are still several issues that need further attention.

1. "the new signatures are in good agreement with the previous, with >50% overlap of signature genes for all clusters."

I am not sure whether "good agreement" is the proper wording here, given that the actual values are all below 65%: 16/25 (64%), 14/27 (51%), 10/17 (58%) for cluster 1, 2, and 3. It gives an idea on the dependence of the methodology on the underlying interaction data, which should be discussed in the main text.

2. "Regarding the results in Table 5, it is hard to perform an enrichment analysis of genes annotated in ClinicalTrials.org, since the information on the signature genes could not be obtained by automatic query of the database (that is actually not structured for such analyses), thus it would be infeasible to manually recover such information for the hundreds of genes in our full network."

I understand that the CT can not be systematically queried by genes but it can be queried by interventions (drugs). Accordingly, one could, in principle, identify the drugs that target the signature genes (e.g., from DrugBank and DGIdb) and check the over-representation of those drugs in CTs compared to all drugs tested in all CTs.

3. In relation to my previous comment (#3) "To strengthen the general applicability of the method and ensure that the authors do not over-fit on existing data, it would definitely help to see whether survival curves are significantly different when a cross-validation approach is used...", I feel that the authors did not address the possibility of over-fitting.

4. I find the analysis on the proximity of signature genes to the mutated genes very interesting. Maybe the authors could elaborate why the cluster 3 did not show significant proximity? It would also be useful to have P-values in regards to the extremeness of the observation on the overlap of signature genes and mutated genes (i.e., the expected probability of covering X signature genes among Y, when one chooses N genes from Ontocancro-COSMIC).

5. Similarly, with regards to my earlier comment (#4), it would be great to see a systematic prioritization of all targets based on their analysis. I believe this is especially pertinent as the authors emphasize that their main aim is suggesting novel targeting strategies / repurposing drugs via the presented approach. Accordingly, among all targets (or drugs binding to these targets) in DrugBank / DGIdb, could the authors propose a scoring system that would allow selection of candidates? How many of the drugs would they call a good candidate at a certain threshold and for how many of them they could find experimental support or evidence in the literature. What would be the expected false positive rate? The authors can also consult to the drug sensitivity & combination data sets provided in recent DREAM challenges for a systematic analysis.

6. The following statements need revision

"Since the signature genes are the most central nodes in each cluster, we hypothesized that they might be suitable drug targets." "Centrality" in the "cluster" causes confusion, as the authors both call the clusters containing samples obtained by gene expression similarity and the protein interaction network involving co-expressed genes as clusters.

"We observed a high overlap (>50%) of our signatures with those obtained by adding to our initial PPI further protein interactions found through different experimental approaches"

"false correlations"

"clustering resulted quite robust"

"only 591 were non-isolated nodes" (~connected to each other or reverse the statement by giving the number of unconnected nodes)

"three cluster-related networks differ in their weight values or for missing links (due to negative z-scores set to zero)."

7. I have had problems with checking the supplementary excel table containing 18 tabs, I think it would be better to separate some of these sheets as new files.

Reviewer #1 (Remarks to the Author):

I very much appreciate both the chance to review this revised paper, the authors' detailed response to the previous reviews, and the substantive changes that the authors made to the manuscript.

I continue to think that this paper addresses a significant challenge in cancer genomics, and I feel that the revisions have cleared up several of the communications problems that contributed to my unfavourable previous review.

Nevertheless, I'm afraid I remain unconvinced of the main assertion in the paper that the method of network clustering and subsequent network analysis that the authors describe contributes significantly to our understanding of clinical behaviour. This assertion rests on two key experiments: the discovery that the expression level of genes comprising the Cluster 2 signature are prognostic, and the sensitivity of two cell lines derived from tumour types in Cluster 2 to a pair of drugs that target Cluster 2 signature genes or processes.

Compared to the original version, the prognostic study of survival is better explained and more strongly supported. However, the right hand panel of Figure 5 discloses a serious confounding issue. Among the tumour types that account for the majority of the tumours analysed, the "bad prognosis" signature is present in roughly 80% of GBM cases, in 40% of BRCA cases, and about 20% of OV cases. The 5-year survival time for GBM is dismal, about 10%, whereas breast cancer and ovarian cancer have much better prognoses.

1) The authors made a good start with the multivariate analysis of prognostic factors (Supplementary Table 5) which showed that the signature remained significant even after regressing out the tumour type, but they may have missed the fact that within the TCGA data set for each tumour type there are differences in tumour stage and grade, which are strong clinical prognostic factors. To make a convincing case, the authors need to perform the survival analysis on each tumour type separately, and to perform a covariate analysis that at least takes into account tumour stage.

ANSWER: we performed the analysis as requested by the reviewer (Survival Analysis Part II: Multivariate data analysis – an introduction to concepts and methods, M J Bradburn, et al., British Journal of Cancer 89, 431–436 2003, 10.1038/sj.bjc.6601119), by considering separately each tumour in Cluster 2 and assuming as covariates the stage/grade of tumours for which these annotations are available in the TCGA database (LUSC, OV, LUAD, UCEC, BRCA). The analysis showed a significant contribution of our signature for LUSC (154 samples, 8 different stages) and a borderline significance for OV ($p = 0.08$). However, in our opinion this further stratification reduces significantly the power of the applied test. For two tumours (LUAD, UCEC) the number of samples is very low, given the stratification into multiple stage (31 and 54 total samples respectively, with single tumour stage groups comprising at most 20 samples, but also as few as 1-2 samples each). The remaining tumours (BRCA, OV) have a large number of samples (>500 each) but they are distributed very unevenly inside the tumour stage groups: in BRCA almost 85% of the samples are in one stage, while one significant stage contains only 1 sample. The idea proposed by the reviewer, that a further stratification should add more information to our analysis, is in principle correct, but the analysis performed shows that the real distribution of the available data does not allow to fully explore this idea. We add the results of these analyses to the text for the reviewer. We can anyway add these last analyses in the Supplementary Material, if the reviewer thinks to be the case.

Cox Multivariate Analysis considering stage, age and clustering as covariates.

(Note: the values for the first group are not shown since it represents the “reference” onto which the other regression coefficient are calculated)

z = Wald statistic value. It evaluates if the regression coefficient of a given variable is significantly nonzero.

P = p-value for significant covariate

Num = number of samples per group (empty = all samples)

LUSC - 154 samples

Covariate	z	P	Num
StagelA			23
StagelB	-0.014	0.98857	61
StagellA	1.84	0.06578	7
StagellB	1.457	0.14519	27
StagelllA	1.054	0.29204	19
StagelllB	1.555	0.12	13
StagelV	-0.117	0.90676	4
age_initial_diagnosis	3.168	0.00153	
kmeans_cluster	-2.335	0.01954	

OV - 589 samples

Covariate	z	p	Num
stageG1			6
stageG2	1,69	0,09	78
stageG3	6,22	4,7E-10	490
stageG4	0,99	0,32	1
stageGB	0,61	0,54	1
stageGX	1,46	0,14	9
age_initial_diagnosis	5,01	5,5E-07	
kmeans_cluster	-1,72	0,08	

LUAD - 31 samples

Covariate	z	p	Num
Stage IA			12
Stage IB	0.736	0.462	11
Stage IIB	0.000	1.000	3
Stage IIIA	0.001	0.999	3
Stage IV	0.000	1.000	1
kmeans_cluster	-0.001	0.999	
age_initial_diagnosis	-1.065	0.287	

BRCA - 567 samples

Covariate	z	p	Num
Stagel			63
StagelA	-3,02	0,0025	37
StagelB	-0,27	0,782	3
Stagell	-2,63	0,008	1
StagellA	-0,38	0,700	189
StagellB	-1,16	0,24	122
StagellIA	0,146	0,88	84
StagellIB	-0,37	0,7	12
StagellIC	1,79	0,07	22
StagelIV	1,21	0,22	14
StageX	0,41	0,67	12
age_initial_diagnosis	2,1	0,03	
kmeans_cluster	-0,2	0,83	

UCEC - 54 samples

Covariate	z	p	Num
stageG1			16
stageG2	0,0016	0,99	9
stageG3	0,0016	0,99	29
age_initial_diagnosis	1,13	0,25	
kmeans_cluster	0,81	0,41	

2) Compared to the original version, the drug sensitivity experiment has not significantly changed. I still find this study of two cell lines across three drugs to be woefully inadequate to support the striking claim that the network analysis method identifies novel drug targeting strategies. The problem is that there is no negative control. As a reader, I cannot tell what is the expected rate of success for any two randomly chosen cell lines and three randomly selected targeted small molecule agents. To do a convincing validation experiment without actually designing a novel therapeutic (which I agree is outside the scope of this paper!), I would suggest that the authors go to one of the several published drug sensitivity databases in which a variety of cell lines have been systematically screened with large number of targeted and conventional agents, and show that the observed rate of positive drug sensitivity results involving predicted pairs of targeted agents and cell lines derived from particular tumour types exceeds what would be predicted by chance. There are several such databases that spring to mind, including the well-studied NCI-60 cell lines, the Sanger Institute's Genomics of Drug Sensitivity in Cancer project (<http://www.cancerrxgene.org/>), and the Broad Institute's Cancer Therapeutics Response Portal (<https://portals.broadinstitute.org/ctrp/>)

ANSWER: we extracted data from the Genomics of Drug Sensitivity in Cancer (GDSC) database to perform several analyses. First, we estimated the percentiles of the IC50 values for the drugs we tested experimentally in comparison to: 1) all drug-cell line combinations found in the database (224,510 drug-cell line pairs obtained from 265 drugs and 1074 cell lines); 2) all drugs specifically tested onto MCF-7 and TG98 cell lines (217 and 216 drugs tested in MCF-7 and T98G cell lines, respectively). In both cases, drug concentrations used in our experiments result much lower than those found in the database, appearing in the lowest percentiles of the distributions. In particular for the two cell lines, our drugs (Bortezomib and BI627) appear in the lower left tail of the distribution, ranging from 0th in the best case to 22th percentile in the worst case.

We also asked if the cell lines in each tumor cluster have lower IC50 values when treated with drugs targeting signature genes in comparison to all other drugs. We manually identified 53, 218, and 49 cell lines that can be associated to the tumours belonging to cluster 1, 2, and 3 respectively (based on TCGA classification, Supplementary Table 23). Moreover, we found 7, 15, and 7 drugs that target genes found in the signatures 1, 2, and 3, respectively. Also in

this case we compared the distribution of IC50 values in response to drugs associated to our clusters with the other drugs found in the database: we observed 4, 18, and 10 cell lines for the clusters 1, 2, and 3, respectively, in which drugs targeting genes in the signatures resulted in significantly lower IC50 values than other drugs. No significant difference in IC50 was associated to higher values of our target drugs as compared with the rest of the drugs, confirming the goodness of our gene signatures. Moreover, we observe a coherent global trend that agrees with our results: cells from the clusters that are treated with drugs targeting genes in the related signatures show lower IC50 values than cells treated with other drugs.

We thank the reviewer for this suggestion, since in our opinion it has increased the robustness of our validation with a wider benchmarking on publicly available data.

We have thus added some of the results in the main text to support our validations, and also added a more detailed description of these analyses in an additional Section of the Supplementary material.

Reviewer #2 (Remarks to the Author):

I thank the authors for their effort on addressing my concerns. In my opinion, there are still several issues that need further attention.

1. “the new signatures are in good agreement with the previous, with >50% overlap of signature genes for all clusters.”

I am not sure whether “good agreement” is the proper wording here, given that the actual values are all below 65%: 16/25 (64%), 14/27 (51%), 10/17 (58%) for cluster 1, 2, and 3. It gives an idea on the dependence of the methodology on the underlying interaction data, which should be discussed in the main text.

ANSWER: We agree with the reviewer that the choice of PPI network is a crucial and still unsolved question: different authors tried to overcome this issue with different strategies tailored to their specific analysis scopes. In this second round of reviewing we have performed many analyses, with a special focus on the validation of our signature over public datasets of clinical trials and drug testing over cell lines, in addition to the experimental validation we performed in vitro. The new results seem to confirm that our signature genes perform significantly better than a random selection from our dataset or in comparison with public data (as in terms of the IC50 of drugs targeting signature genes in several comparisons with >200000 in vitro experiments), thus we remain quite confident of our analysis strategy and choice of data and network architecture, but this does not exclude that even better results could be achieved with an “optimal” PPI. We have added a comment on this topic and the relevance of our results in the conclusive section of the paper (with the addition of recent references dealing with this topic), but we think that an exhaustive description of this topic goes far beyond the scope of this paper. To the referee's knowledge, we are performing a study on a large set of PPI network taken from literature trying to answer to this issues, but it is still in a draft version, and we don't know if it will be available on a public archive before the resubmission of this paper.

We have rephrased our statement, restating it in a more neutral manner in the conclusive section of the paper, and also added comments and references on actual literature dedicated to this topic.

2. “Regarding the results in Table 5, it is hard to perform an enrichment analysis of genes annotated in ClinicalTrials.org, since the information on the signature genes could not be obtained by automatic query of the database (that is actually not structured for such analyses), thus it would be infeasible to manually recover such information for the hundreds of genes in our full network”. I understand that the CT cannot be systematically queried by genes but it can be queried by interventions (drugs). Accordingly, one could, in principle, identify the drugs that target the signature genes (e.g., from DrugBank and DGldb) and check the over-representation of those drugs in CTs compared to all drugs tested in all CTs.

ANSWER: We evaluated all studies in ClinicalTrials.gov that specifically addressed oncologic conditions, retrieving the respective drug-gene interactions from the Drug-Gene Interaction Database (DGldb). We estimated the overrepresentation, in the ClinicalTrials.gov, of drugs targeting genes in the signature in relation to: 1) drugs targeting all genes in the BioPlex network, and 2) drugs targeting genes in the BioPlex-Ontocancro network.

In the first case the overrepresentation was enormously significant for all signatures. In the second case, the overrepresentation was significant for cluster 2 and in the border line of significance for cluster 3 ($p=0.08$, but it would drop to $p=0.045$ if the drugs were 7 instead of 6,

see the dedicated Supplementary section), even if we remark that although the cluster 1 was not significant ($p=0.21$) the trend was correct (an almost double odds ratio of drugs targeting signature genes in as compared to the Bioplex-OntoCancro Network). We added a supplementary section to describe this analysis, referring to it in the main text when discussing the validation of our signatures.

3. In relation to my previous comment (#3) "To strengthen the general applicability of the method and ensure that the authors do not over-fit on existing data, it would definitely help to see whether survival curves are significantly different when a cross-validation approach is used...", I feel that the authors did not address the possibility of over-fitting.

ANSWER: in response to the reviewer's request, we remark that our signatures were chosen by ranking genes via a centrality measure applied on the networks related to the 3 clusters, so in principle there is no relation with a putative "discriminant power" of these signatures in a survival analysis. Anyway, to overcome the possibility of an excessive dependence of our signature on the dataset, we recalculated the signatures for 100 times, by applying the whole analysis pipeline (correlation matrix construction, filtering through CLR, centrality measurement and selection of top 10% genes) over a 50% subsampling of the whole sample dataset for each cluster. We found that most of the signature genes are conserved in these subsampling, confirming the robustness of our signatures to perturbations in sample space. The results of this procedure are described in an additional Supplementary Section, and have been commented in the text.

4. I find the analysis on the proximity of signature genes to the mutated genes very interesting. Maybe the authors could elaborate why the cluster 3 did not show significant proximity? It would also be useful to have P-values in regards to the extremeness of the observation on the overlap of signature genes and mutated genes (i.e., the expected probability of covering X signature genes among Y, when one chooses N genes from Ontocancro-COSMIC).

ANSWER: we performed several structural analyses of the cluster-specific networks, keeping into account the information on mutated and signature genes. The networks of the three clusters seem very similar from a topological point of view, the only difference that could explain the different behaviour of cluster 3 is that the signature genes (measured with Cohen's d score) result more close to the network core, while mutant genes seem closer to the network periphery, and this could increase the average distance between them. We expanded the supplementary section with the new results and referenced in the the main text.

Moreover, we extracted from the COSMIC dataset the complete list of mutated genes that 1) can be associated to our tumours and 2) appear in our Bioplex-Ontocancro network. This list counts 105 genes, and its overlap with the genes in our signature is very small: 3, 2, and 1 genes in common with the 3 cluster signatures, respectively (See Supplementary). The maximum overlap is thus very small, and it is comparable with the overlap we observed with the selected COSMIC mutated gene list, consisting of 2, 2, and 1 genes for the 3 clusters, respectively (see the extension of the Supplementary Section "Overlap and Distance between signature genes and mutated genes").

5. Similarly, with regards to my earlier comment (#4), it would be great to see a systematic prioritization of all targets based on their analysis. I believe this is especially pertinent as the authors emphasize that their main aim is suggesting novel targeting strategies / repurposing drugs via the presented approach. Accordingly, among all targets (or drugs binding to these targets) in DrugBank/DGIdb, could the authors propose a scoring system that would allow selection of candidates? How many of the drugs would they call a good candidate at a certain threshold, and for how many of them they could find experimental support or evidence in the literature. What would be the expected false positive rate?

The authors can also consult to the drug sensitivity & combination data sets provided in recent DREAM challenges for a systematic analysis.

ANSWER: as we stated previously, we believe it is more robust to consider the top-ranking genes (first decile) sorted by Spectral Centrality as equally important, but in order to have a gene-by-gene prioritization of our target we have added the whole ranked list of genes for each cluster as an Excel File in the Supplementary Material.

With regard to the second part of the question, we have added comments in the main text and a full extra section of Supplementary Materials regarding the validation of our signature in public databases of cell lines and drug treatments (Genomics of Drug Sensitivity in Cancer - GDSC - consisting of 224,510 drug-cell line pairs obtained from 265 drugs and 1074 cell lines). These new results extend our in vitro validation over a small number of drugs and cell lines, and confirm the better performance of our drug targets (in terms of IC50) as compared to different drugs applied on the same cell types we tested experimentally, but also with respect to all cell lines in the database that can be associated to the tumours in our tumour clusters.

MINOR

6. The following statements need revision

“Since the signature genes are the most central nodes in each cluster, we hypothesized that they might be suitable drug targets.” “Centrality” in the “cluster” causes confusion, as the authors both call the clusters containing samples obtained by gene expression similarity and the protein interaction network involving co-expressed genes as clusters.

“We observed a high overlap (>50%) of our signatures with those obtained by adding to our initial PPI further protein interactions found through different experimental approaches“

“false correlations”

“clustering resulted quite robust”

“only 591 were non-isolated nodes” (~connected to each other or reverse the statement by giving the number of unconnected nodes)

“three cluster-related networks differ in their weight values or for missing links (due to negative z-scores set to zero).”

ANSWER: we rephrased the sentences following the suggestions of the reviewer

7. I have had problems with checking the supplementary excel table containing 18 tabs, I think it would be better to separate some of these sheets as new files.

ANSWER: we divided the original Excel Supplementary file into more files, as requested by the reviewer.

Reviewers' comments:

Reviewer #1 (Remarks to the Author):

Thank you for the opportunity to re-review this manuscript and for the authors' responsiveness to my earlier concerns. In my previous review, I had raised two main concerns. The first was that the survival analysis might be confounded by the intermixing of multiple tumour types, grades and stages; the second was that there was no "negative control" to evaluate the observed drug response in the cell lines and drugs against random expectation.

With respect to the second concern, I am very happy to see that the authors performed the suggested comparison of IC50s using the online databases of drug response. I believe this does satisfy my earlier concerns and greatly strengthens the authors' case that the network-based signatures can predict drug response.

With respect to the first concern, I note that the cluster 2 signature does survive stratification by tumour type and age, but that further stratification by stage led to a reduction in significance of the signature in all tumour types, and a loss of significance in breast, lung adeno, and uterine cancer. This is a common problem in biomarker discovery, and may be the result of late tumour evolution that characterises the more advanced stages. I'm inclined not to make a big deal about this. After all, you can turn the results around and say that the cluster 2 signature is a good predictor of tumour stage. It still shows that there is a correlation between the network-based signature and a clinical parameter - just not an independent prediction of survival. However, I would ask that the authors add the multivariate analysis to the supplementary material and mention the results in the main text.

I am still concerned about the presentation of Figure 5, in which the overall survival curves of six different tumour types are merged. I do think this is a misleading figure and suggest that separate survival curves be drawn for each tumour type.

Reviewer #2 (Remarks to the Author):

In the earlier versions of the manuscript, I had been concerned on the robustness of the method to identify potential therapeutic targets and dependence of the findings (three clusters & signature genes) with respect to the underlying data sets such as the initial cancer gene list, protein interactions, number of clusters and signature genes. The authors have addressed this issue to a considerable extent and now provide evidence on the potential impact of such discrepancy in the data sets. In my opinion, there are several issues that need to be clarified further.

1. "We found that most of the signature genes are conserved in these subsampling: 25/25 (100%) signature genes of cluster 1 were found at least one time in the resampled signatures; 23/27 (85%) in cluster 2; 17/17 (100%) in cluster 3 (see Table below, clusters 1-3 from left to right, with gene name and count of the appearances during subsampling). These results confirm the robustness of our signatures to perturbations in sample space." ... ' We found that most of the signature genes are conserved in these subsampling, confirming the robustness of our signatures to perturbations in sample space."

Supplementary Table 4 shows that most of the signature genes do not appear half of the times when different subsets of initial gene list are used. Given only half of the initial gene list is used in each repetition, this is somewhat expected but also pointing to the importance of the initial selection of the genes. In my opinion, the current interpretation of authors' on the signatures being robust against perturbations is not representing this result / limitation, which should be highlighted in the text.

2. Similar to my previous comment, the study relies on the initial clustering of gene expression

across tumors. The authors use 3 clusters based on the hierarchical clustering somewhat arbitrarily. Some insights on the choice of the number of clusters would be helpful.

3. "The comparison was performed using the Fisher's exact test. By considering all drugs that target genes present in the BioPlex network, the drugs targeting genes in all three signatures were significantly enriched in ClinicalTrials.gov (Supplementary Table 7). By considering all drugs that target genes present in the BioPlex-Ontocancro network, the enrichment was significant for drugs targeting genes in cluster 2 signature ($p = 0.0015$) and in the border line of significance for those targeting genes in cluster 3 ($p = 0.085$, Supplementary Table 7). Due to the small number of samples, we remark that for cluster 3 just one more drug counted in the "Drug targeting signature genes in clinical trials" (i.e. 7 vs 123) would have led to a significant difference ($p=0.045$). Anyway the trend is what would be expected for all three clusters (at least a double odds ratio in the worst case of cluster 1)."

The comparison using the whole BioPlex network is a bit confusing to me. Wouldn't the signature genes be different (than those they obtained on BioPlex-Ontocancro network), had the authors used this network in their analyzes (and thus the P-values they obtained would be different)? On the other hand, the lack of enrichment of drugs in cluster 1 & 3 weakens the argument that the argument on "Our study thus provides a list of genes and pathways with the potential to be used, singularly or in combination, for the design of novel treatment strategies" and should potentially be revised in the text.

4. "We observed that the effect of the gene signature on dividing patients remained prevailing even when considering confounding factors such age and tumor type (see Supplementary Material"

Given the correlation across genes, random gene sets have been shown to capture transcriptomic differences across groups of patients (see Pubmed PMIDs 29153835 & 22028643). Accordingly, the segregation between patients provide little value toward the clinical value of these signatures, which again should be discussed along with the mentioned references.

5. Are the P-values from GO Enrichment analysis corrected for multiple hypothesis testing?

6. "Since the signature genes are the most central nodes in the three tumor cluster networks, we hypothesized that they might be suitable drug targets."

This could be better motivated by referring to previous works on the drug targets being more central in the human interactome (see Pubmed PMIDs 23384594 & 26831545).

7. "We also asked whether signature genes in each cluster could predict a better response of cell lines to the related drugs when compared with all drugs tested in GDSC. We identified 56, 218, and 49 cell lines associated to the tumour types grouped in cluster 1, 2, and 3, respectively (according to TCGA classification, Supplementary Table 23). We identified 7, 15, and 7 drugs targeting genes belonging to signature 1, 2, and 3, respectively. We observed that most of the cell lines in each cluster (51/53, 103/218, and 47/49, respectively) were more sensitive to drugs targeting signature genes as compared with all other drugs (according to IC50 values, Supplementary Figures 17-19). Specifically, we found that 4, 18, and 10 cell lines in clusters 1, 2, and 3, respectively, showed significant differences ($p<0.05$, Student's T test, Figure 8), all of them displaying lower IC50 value when treated with drugs targeting signature genes."

I find this analysis quite interesting and potentially supporting the use of the identified signature genes toward developing novel therapies. In addition to the current comparison, a comparison between drugs that are targeted by signature genes vs drugs that are targeted by non-signature genes in the subnetwork (rather than any other gene) would help position the results better toward the potential of signature genes as therapeutic targets.

Consider revising the following:

- "By superimposing the gene-gene correlation matrices (calculated with the samples of all tumours inside each cluster) onto the BioPlex-Ontocancro network (common to all tumours) we obtained three weighted networks, each with approximately 80% nodes and 60% edges of the

original BioPlex-Ontocancro network (Table 2, see Supplementary Figures 1-4). " First time the authors talk about the "original BioPlex-Ontocancro network", a brief description of this network would help. Also consider removing "original".

- "We observed an overlap (>50%) of our signatures with those obtained by adding to our initial PPI further protein interactions found through different experimental approaches^{26,27} (i.e. yeast-two hybrid), supporting their robustness and biological relevance (Supplementary Table 3)."

Unclear

- "We remark that the chosen signatures have only a small overlap with the most central nodes of the original "full" Bioplex-Ontocancro network not filtered by the cluster-specific correlation matrices (3/25, 13/27 and 4/24 common genes for clusters 1, 2, 3, respectively) showing how the information on gene expression profile is highly specific for the considered tumour clusters"

Unclear

- Anova & ANOVA inconsistent case throughout text

- The alpha cutoff used for different statistical tests conducted in the study should be mentioned explicitly

NOTE: all modified parts in the text (both Main and Supplementary) are highlighted in red for a better identification.

Reviewers' comments:

Reviewer #1 (Remarks to the Author):

Thank you for the opportunity to re-review this manuscript and for the authors' responsiveness to my earlier concerns. In my previous review, I had raised two main concerns. The first was that the survival analysis might be confounded by the intermixing of multiple tumour types, grades and stages; the second was that there was no "negative control" to evaluate the observed drug response in the cell lines and drugs against random expectation.

With respect to the second concern, I am very happy to see that the authors performed the suggested comparison of IC50s using the online databases of drug response.

I believe this does satisfy my earlier concerns and greatly strengthens the authors' case that the network-based signatures can predict drug response.

With respect to the first concern, I note that the cluster 2 signature does survive stratification by tumour type and age, but that further stratification by stage led to a reduction in significance of the signature in all tumour types, and a loss of significance in breast, lung adeno, and uterine cancer. This is a common problem in biomarker discovery, and may be the result of late tumour evolution that characterises the more advanced stages. I'm inclined not to make a big deal about this. After all, you can turn the results around and say that the cluster 2 signature is a good predictor of tumour stage. It still shows that there is a correlation between the network-based signature and a clinical parameter - just not an independent prediction of survival. However, I would ask that the authors add the multivariate analysis to the supplementary material and mention the results in the main text.

I am still concerned about the presentation of Figure 5, in which the overall survival curves of six different tumour types are merged. I do think this is a misleading figure and suggest that separate survival curves be drawn for each tumour type.

A: we added the multivariate analysis to the Supplementary Section, and also commented in the main text the different results obtained when separating the different tumours survival curves. Moreover, we added to Figure 5 boxes representing the separate survival curves mentioned in the text for each single tumour type.

Reviewer #2 (Remarks to the Author):

In the earlier versions of the manuscript, I had been concerned on the robustness of the method to identify potential therapeutic targets and dependence of the findings (three clusters & signature genes) with respect to the underlying data sets such as the initial cancer gene list, protein interactions, number of clusters and signature genes. The authors have addressed this issue to a considerable extent and now provide evidence on the potential impact of such discrepancy in the data sets. In my opinion, there are several issues that need to be clarified further.

1. “We found that most of the signature genes are conserved in these subsampling: 25/25 (100%) signature genes of cluster 1 were found at least one time in the resampled signatures; 23/27 (85%) in cluster 2; 17/17 (100%) in cluster 3 (see Table below, clusters 1-3 from left to right, with gene name and count of the appearances during subsampling). These results confirm the robustness of our signatures to perturbations in sample space.” ... ‘ We found that most of the signature genes are conserved in these subsampling, confirming the robustness of our signatures to perturbations in sample space.’”

Supplementary Table 4 shows that most of the signature genes do not appear half of the times when different subsets of initial gene list are used. Given only half of the initial gene list is used in each repetition, this is somewhat expected but also pointing to the importance of the initial selection of the genes. In my opinion, the current interpretation of authors’ on the signatures being robust against perturbations is not representing this result / limitation, which should be highlighted in the text.

A: in order to test the robustness of our signature, we applied a bootstrapping technique performing the whole analysis pipeline (namely: correlation between expression profiles, mapping onto PPI-ontocancro network, signature identification through 90th percentile of Spectral Centrality) to a subset of the available samples (only 50% of the total patients selected every time, and not 50% of the genes as pointed out by the referee) for 100 different subsamplings, which in our opinion is a strong perturbation of the initial patient dataset (at each simulation the available dataset is reduced by 50%). We better clarified our bootstrapping results in the text (Methods Section) and added our comments as required (in Supplementary Section).

2. Similar to my previous comment, the study relies on the initial clustering of gene expression across tumors. The authors use 3 clusters based on the hierarchical clustering somewhat arbitrarily. Some insights on the choice of the number of clusters would be helpful.

A: We forgot to mention in the previous versions of the paper (and we thank the reviewer for this) that the clusters were selected by applying the Dynamic Tree Cut algorithm. We have added the description of the clustering technique in the Materials and Methods Section accordingly.

3. “The comparison was performed using the Fisher’s exact test. By considering all drugs that target genes present in the BioPlex network, the drugs targeting genes in all three signatures were significantly enriched in ClinicalTrials.gov (Supplementary Table 7). By considering all drugs that target genes present in the BioPlex-Ontocancro network, the enrichment was significant for drugs targeting genes in cluster 2 signature ($p = 0.0015$) and in the border line of significance for those targeting genes in cluster 3 ($p = 0.085$, Supplementary Table 7). Due to the small number of samples,

we remark that for cluster 3 just one more drug counted in the "Drug targeting signature genes in clinical trials" (i.e. 7 vs 123) would have led to a significant difference ($p=0.045$). Anyway the trend is what would be expected for all three clusters (at least a double odds ratio in the worst case of cluster 1)."

The comparison using the whole BioPlex network is a bit confusing to me. Wouldn't the signature genes be different (than those they obtained on BioPlex-Ontocancro network), had the authors used this network in their analyzes (and thus the P-values they obtained would be different)? On the other hand, the lack of enrichment of drugs in cluster 1 & 3 weakens the argument that the argument on "Our study thus provides a list of genes and pathways with the potential to be used, singularly or in combination, for the design of novel treatment strategies" and should potentially be revised in the text.

A: Following referee's suggestions, we removed the analysis on the full Bioplex network from the main text, and kept only in the Supplementary Section.

Regarding the enrichment of signature genes in ClinicalTrials, we make the following remark: since we are looking for possibly novel targets (or known targets in some tumours that were not considered in other tumours) it is plausible that not all of them are already object of existing clinical trials. Cluster two is the one with most tumours, so the probability of having more clinical trials ongoing could have been higher: the significant results obtained for cluster two and the correct trends in the ratios for cluster 1 and 3 all point in the same direction which is the expected one. We have rephrased the sentence in the Abstract section trying to emphasize the potentiality of our approach.

4. "We observed that the effect of the gene signature on dividing patients remained prevailing even when considering confounding factors such age and tumor type (see Supplementary Material" Given the correlation across genes, random gene sets have been shown to capture transcriptomic differences across groups of patients (see Pubmed PMIDs 29153835 & 22028643).

Accordingly, the segregation between patients provide little value toward the clinical value of these signatures, which again should be discussed along with the mentioned references.

A: We tested our signature on a survival analysis because it is very much used in clinical studies, and it is easily understandable by clinicians. The proposed references are very interesting, but we remark that we already tested our global survival analysis against 1000 random signatures in the previous revision, in order to better understand and control the role of randomness in the signature. Given the proposed comments and references, we included a remark in the result session regarding the results for the different tumours considered separately, and added the references (ref. 52, 53) together with a comment in the final Discussion Section.

5. Are the P-values from GO Enrichment analysis corrected for multiple hypothesis testing?

A: Yes they are, the corrections to the text were made where necessary (i.e. in the supplementary tables).

6. "Since the signature genes are the most central nodes in the three tumor cluster networks, we hypothesized that they might be suitable drug targets." This could be better motivated by referring to previous works on the drug targets being more central in the human interactome (see Pubmed PMIDs 23384594 & 26831545).

A: we have added the proposed references (ref. 25,26) and commented them in the Main text.

7. “We also asked whether signature genes in each cluster could predict a better response of cell lines to the related drugs when compared with all drugs tested in GDSC. We identified 56, 218, and 49 cell lines associated to the tumour types grouped in cluster 1, 2, and 3, respectively (according to TCGA classification, Supplementary Table 23). We identified 7, 15, and 7 drugs targeting genes belonging to signature 1, 2, and 3, respectively. We observed that most of the cell lines in each cluster (51/53, 103/218, and 47/49, respectively) were more sensitive to drugs targeting signature genes as compared with all other drugs (according to IC50 values, Supplementary Figures 17-19). Specifically, we found that 4, 18, and 10 cell lines in clusters 1, 2, and 3, respectively, showed significant differences ($p < 0.05$, Student's T test, Figure 8), all of them displaying lower IC50 value when treated with drugs targeting signature genes.”

I find this analysis quite interesting and potentially supporting the use of the identified signature genes toward developing novel therapies. In addition to the current comparison, a comparison between drugs that are targeted by signature genes vs drugs that are targeted by non-signature genes in the subnetwork (rather than any other gene) would help position the results better toward the potential of signature genes as therapeutic targets.

A: According to referee's request we have performed, in addition to previous analysis, the comparison between drugs that target signature genes vs drugs that target non-signature genes found only in the BioPlex-Ontocancro network, adding it to the text close to the previous results.

Consider revising the following:

- “By superimposing the gene-gene correlation matrices (calculated with the samples of all tumours inside each cluster) onto the BioPlex-Ontocancro network (common to all tumours) we obtained three weighted networks, each with approximately 80% nodes and 60% edges of the original BioPlex-Ontocancro network (Table 2, see Supplementary Figures 1-4).” First time the authors talk about the “original BioPlex-Ontocancro network”, a brief description of this network would help. Also consider removing “original”.

A: we added a brief description in the main text as requested (the complete description of the Ontocancro database is in the Methods section)

- “We observed an overlap (>50%) of our signatures with those obtained by adding to our initial PPI further protein interactions found through different experimental approaches 26,27 (i.e. yeast-two hybrid), supporting their robustness and biological relevance (Supplementary Table 3).” Unclear

A: we restated the sentence making it more clear

- “We remark that the chosen signatures have only a small overlap with the most central nodes of the original “full” Bioplex-Ontocancro network not filtered by the cluster-specific correlation matrices (3/25, 13/27 and 4/24 common genes for clusters 1, 2, 3, respectively) showing how the information on gene expression profile is highly specific for the considered tumour clusters” Unclear

A: we restated the sentence with a better clarification of the details

- Anova & ANOVA inconsistent case throughout text

A: we corrected with the uppercase ANOVA along the text

- The alpha cutoff used for different statistical tests conducted in the study should be mentioned explicitly

A: we mentioned the cutoff where requested

REVIEWERS' COMMENTS:

Reviewer #1 (Remarks to the Author):

The authors have satisfactorily addressed all the points raised in my previous reviews and I have no further questions or concerns.

Reviewer #2 (Remarks to the Author):

The authors have addressed my comments.